Journal of Data-centric Machine Learning Research (2023)                    Submitted 12/23; Published -

# ATCO2 corpus: A Large-Scale Dataset for Research on Automatic Speech Recognition and Natural Language Understanding of Air Traffic Control Communications

**Juan Zuluaga-Gomez**[★,1,2]**, Karel Veselý**[3] **Igor Szöke**[3,4] **Alexander Blatt**[5]
**Petr Motlicek**[1,3] **Martin Kocour**[3] **Mickael Rigault**[6] **Khalid Choukri**[6]
**Amrutha Prasad**[1,3] **Seyyed Saeed Sarfjoo**[1] **Iuliia Nigmatulina**[1,7] **Claudia Cevenini**[8]
**Pavel Kolčárek**[9] **Allan Tart**[10] **Jan Černocký**[3] **Dietrich Klakow**[5]

[1]*Speech & Audio Processing Group, Idiap Research Institute, 1920 Martigny, Switzerland*

[2]*LIDIAP, Ecole Polytechnique Fédérale de Lausanne, 1015 Lausanne, Switzerland*

[3]*Faculty of Information Technology, Brno University of Technology, 60190 Brno, Czech Republic*

[4]*ReplayWell, Brno, Czech Republic*

[5]*Saarland University, Saarland Informatics Campus, Germany*

[6]*Evaluations and Language Resources Distribution Agency (ELDA), 75013 Paris, France*

[7]*Institute of Computational Linguistics, University of Zurich, 8050 Zurich, Switzerland*

[8]*Romagna Tech, Forli, Italy*

[9]*Honeywell, Brno, Czech Republic*

[10]*OpenSky Network, 3400 Burgdorf, Switzerland*

**★Corresponding author: juan-pablo.zuluaga@idiap.ch**

**Reviewed on OpenReview:** `https://openreview.net/forum?id=XXXX`

**Editor:** –

## Abstract

Personal assistants, automatic speech recognizers and dialogue understanding systems are becoming more critical in our interconnected digital world. A clear example is air traffic control (ATC) communications. ATC aims at guiding aircraft and controlling the airspace in a safe and optimal manner. These voice-based dialogues are carried between an air traffic controller (ATCO) and pilots via very-high frequency radio channels. In order to incorporate these novel technologies into ATC, large-scale annotated datasets are required to develop the data-driven AI systems. Two examples are automatic speech recognition (ASR) and natural language understanding (NLU). However, ATC is considered a low-resource domain. In this paper, we introduce the *ATCO2 corpus*, a dataset that aims at fostering research on the challenging ATC field, which has lagged behind due to lack of annotated data. In addition, we also open-source a GitHub repository[1] that contains data preparation and training scripts useful to replicate our baselines related to ASR and NLU. The *ATCO2 corpus* covers 1) audio and radar data collection and pre-processing, 2) pseudo-transcriptions of speech audio, and 3) extraction of ATC-related named entities. The *ATCO2 corpus* is split into three subsets: (i) *ATCO2-test-set corpus* contains 4 hours of ATC speech with manual transcripts and a subset with gold transcriptions for named-entity recognition (callsign, command, value) and speaker role detection. (ii) The *ATCO2-test-set-1h corpus* is a one-

---

1. Public GitHub repository `https://github.com/idiap/atco2-corpus`.

hour open-sourced subset from the 4h test set.[2] (iii) The *ATCO2-PL-set corpus* consists of 5'281 hours of pseudo-transcribed ATC speech enriched with contextual information (list of relevant n-gram sequences per utterance), speaker turn information, signal-to-noise ratio estimate and English language detection score per sample. The whole *ATCO2 corpus* is publicly distributed through ELDA catalog.[3] We expect the corpus will foster research on robust ASR and NLU not only in the field of ATC communications but also in the general research community.

**Keywords:** Automatic Speech Recognition, Spoken Language Understanding, Natural Language Processing, Air Traffic Control Communications.

# 1 Introduction

The corpus introduced in this research is within the domain of civil air traffic control (ATC) communications and management. ATC aims at managing the airspace in a safe and optimal manner. The communications are either via spoken or data-link messages, while the time-critical messages are always spoken. Each spoken message involves an air traffic controller (from now on, ATCo) issuing spoken flight instructions to aircraft pilots during all phases of the flight. The dialogue follows a well-defined grammar and set of rules that ensures safety, reliability, and efficiency (ICAO, 2020; Helmke et al., 2018). The overall dialogue setup can be seen as a multi-speaker and multi-turn conversation.

Commonly, an ATCo addresses several pilots in a short period of time, which in turns becomes the main cause of increased workload. This is also a limiting factor to increase already existent systems' capacity, i.e., there is large space for optimization by only reducing ATCo's workload. A significant bottleneck in the pipeline is the significant latency arising from an ATCo issuing a command by voice and inserting it manually into the ATCo's workstation (for control and record). Recent advances in automatic speech recognition (ASR) and natural language processing (NLP) technologies have opened new ways where ATCo's workload can be reduced[4] by integrating different systems in a cascade format. The systems for extracting the actual meaning from the original audio signal are commonly known as spoken language understanding (SLU).

This paper introduces the *ATCO2 Corpus* collected as part of ATCO2 project.[5] ATCO2 developed a platform to collect, organize, pre-process and automatically transcribe ATC dialogues.[6] The main bottleneck towards ASR or natural language understanding (NLU) techniques for ATC is the lack of transcribed data. Further, its collection and transcription requires professional human annotators, thus becoming excessively costly and impractical. This study presents how the entire data collection and transcription process can be efficiently accelerated by using already existing machine learning (ML) concepts.

ATCO2 project has made sizeable progress on independent systems for ATC, such as robust ASR (Zuluaga-Gomez et al., 2020a), NLP (Zuluaga-Gomez et al., 2020b), and

---

2. Free to download, available at: `https://www.atco2.org/data`

3. `https://catalog.elra.info/en-us/repository/browse/ELRA-S0484/`

4. Note that workload reduction might be translated to reduced flight time. Thus decreasing the overall operational costs and the environmental impact of aircraft.

5. Automatic collection and processing of voice data from air-traffic communications, website: `https://www.atco2.org/`.

6. This pipeline can be easily adopted into other applications where data scarcity is a latent problem, but access to unlabeled/non-transcribed data is permissible e.g., patient–physician dialogues.

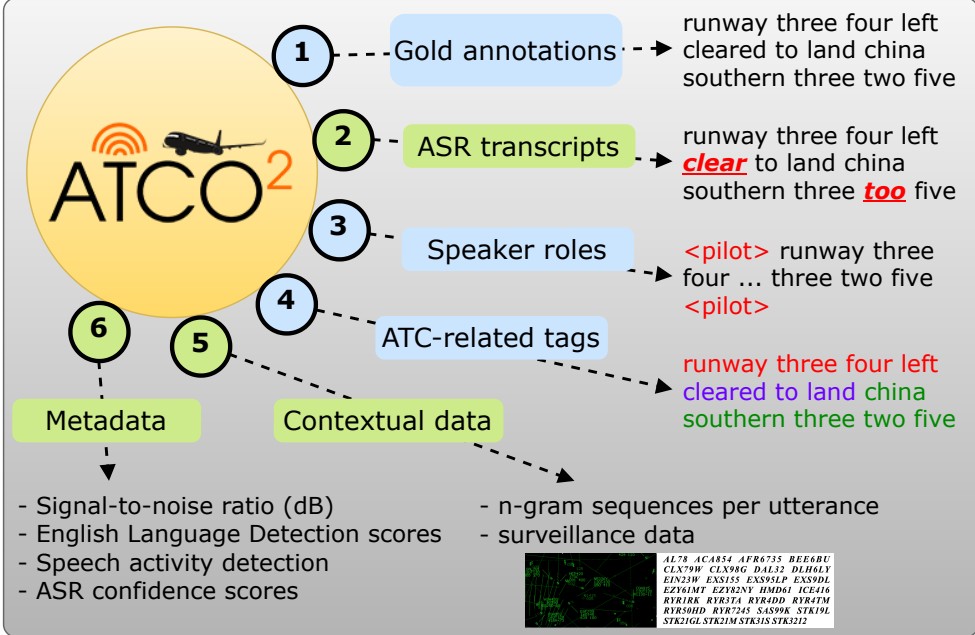

Figure 1: **ATCO2 corpus ecosystem**. Blue circles denote transcriptions only available for *ATCO2 test set corpus.* Green circles denote transcriptions and metadata available for both *ATCO2 test set* corpus and *ATCO2 pseudo-labeled* corpus (see Table 1 bottom).

diarization and segmentation (Kocour et al., 2021b). However, until today, these systems are close to non-existent in real-life ATC operations. In part, this is due to the intrinsic complexity of the task, and mainly to the lack of annotated data (Cordero et al., 2012).

The overall *ATCO2 Corpus* ecosystem is depicted in Figure 1. We release two corpora targeted to ATC for research in robust ASR, NLP and NLU: (i) the *ATCO2-test-set corpus* and (ii) the *ATCO2 pseudo-labeled set corpus* (*ATCO2-PL-set corpus*). The former contains word-level and named entities[7] gold transcriptions. In total, we release 4 hours of speech with useful metadata (see the blue circles in Figure 1). The latter, *ATCO2-PL-set corpus*, contains ∼5'281 hours of pure ATC speech, where each utterance includes a detailed set of metadata. This includes pseudo-transcripts obtained from an in-domain ASR system (including diarization and segmentation information), contextual information (list of word sequences for lattice-boosting of callsigns), signal-to-noise ratio (SNR) estimates, and English language detection (ELD) scores. This is depicted in the green circles in Figure 1. Even though this is not the first publicly available corpus related to ATC communications (Pigeon et al., 2007; Hofbauer et al., 2008; Šmídl et al., 2019b; Godfrey, 1994; Yang et al., 2020), to author's knowledge, this is the first corpus that conveys ATC data with parallel labels for ASR and spoken-based tasks, including named entity recognition (NER), slot filling (SF), and sequence classification.

---

7. Our named entity recognition (NER) classes are: Callsign, Command, Value and Unnamed Phrase. Part of the NER labels can be used to train and test SLU systems for slot filling.

An overview of the data processing pipeline developed by ATCO2 project and used to collect the *ATCO2 corpus* is depicted in Figure 2 (detailed description in §4.3). The data processing pipeline consists of 1) speech pre-processing tools (segmentation, volume adjustment and discarding noisy recordings), 2) diarization (split audio per speaker), 3) ASR, 4) English language detection (ELD), 5) speaker role detection (SRD) e.g., ATCo or pilot, and 6) labeling of callsigns, commands and values with named entity recognition (NER). ATCO2 utilised this pipeline to pre-process the *ATCO2-PL-set corpus* and *ATCO2-test-set corpus*, covering audio data from more than ten airports worldwide. The ATCO2 corpus is publicly available in ELDA catalog: `http://catalog.elra.info/en-us/repository/browse/ELRA-S0484/`. Further details about the data collection and pre-processing are covered in the Appendix A and our previous paper (Kocour et al., 2021b).

The developed pipeline is running live at the Spokendata server.[8]. The data is automatically fed, filtered, and pre-transcribed on a daily basis. We are searching for volunteers, both for feeding data from new airports (see §4.1) or for correcting the automatic transcripts (see §4.2). Overall, the *ATCO2 corpus* can be used to train a robust ASR system for the ATC domain. With the NER transcriptions it is possible to train models for SLU applications, i.e., extraction of meaning from ATC speech. We also believe that the pipeline developed by ATCO2 can partially transfer well to data collection and transcription on different domains, e.g., call-centers conversations, or medical recordings.

**Motivation** Speech and text-based processing tools for ATC data could work better if we had a large amount of reliably annotated data. However, the collection and manual transcription requires qualified personnel, and it is costly. In addition, the recordings are often noisy (SNR below 15 dB), accented or with high speech rate (compared to conversational, read or spontaneous speech). Aligned to solve this, *ATCO2 corpus* answers four big challenges:

1. Current ATC corpora are limited to automatic speech recognition. In fact, ASR is only a small submodule of the whole pipeline and many more downstream tasks are indeed required, e.g., ATC-related NER or callsign detection and extraction. In our case, those are callsigns, commands and their values. *ATCO2 corpora* goes in this line and further by releasing gold transcriptions to train systems on ASR, NER, ELD, and SRD.

2. Research on ATC communications has lagged behind due to the lack of annotated data. The primary rational motive is the high transcription cost. ATC communications require eight to ten man-hours effort (Cordero et al., 2012) to annotate one hour of raw controller-pilot dialogues. Primarily because it requires highly trained participants, often active or retired ATCos. In total, after further pre-processing (e.g., silence and noisy segments removal) around one man-week work yields roughly an hour of transcriptions without silences (Cordero et al., 2012; Ferreiros et al., 2012). This number increases if further metadata is required, e.g., word-level tagging for NER. We address these issues by developing an efficient pipeline to collect, pre-process and automatically annotate ATC data (see Figure 2). Using pre-transcribed data rather than transcribing from scratch reduces drastically the overall transcription process time period. Likewise, our observations during ATCO2 project reveal that the real-time factor (RTF) among the data transcribers varies drastically. For instance, untrained transcribers exhibited up to 50 RTF for transcribing ATC speech, including channel and NER tagging. However, trained transcribers reach as low as 20 RTF for the whole transcription process.

---

8. `https://www.spokendata.com/atco2`

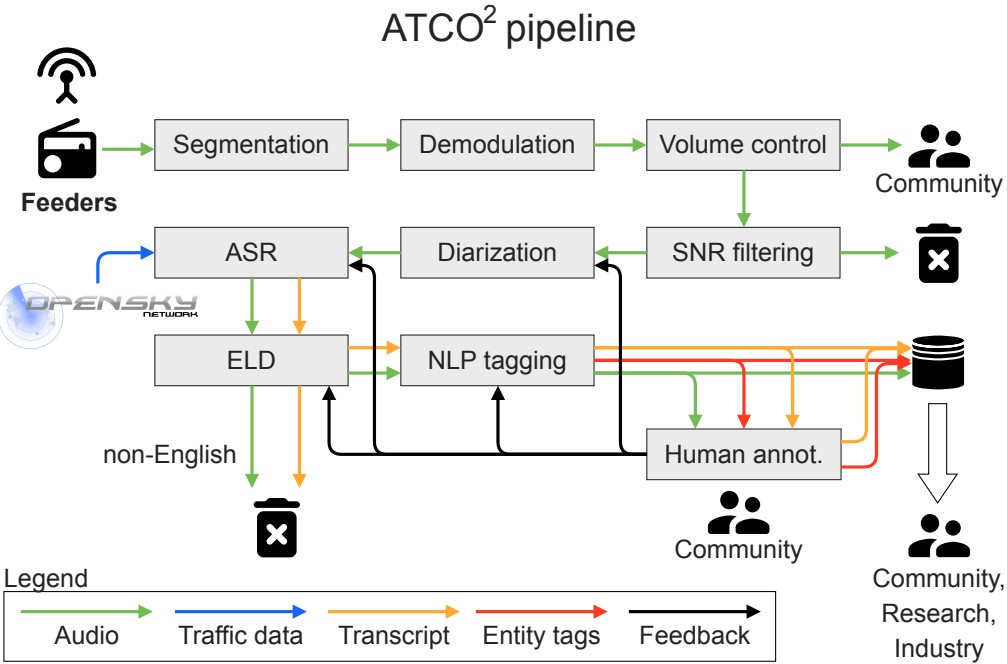

Figure 2: Data collection and data-processing pipeline developed in ATCO2 project. SNR: signal-to-noise; ASR: automatic speech recognition; ELD: English language detection; NLP: natural language processing.

3. Large domain shift between ATC and non-ATC corpora. Current ATC corpora contain data from only a few airports, and some were collected in clean and quiet simulation or training rooms (Pigeon et al., 2007; Hofbauer et al., 2008; Šmídl et al., 2019b). Even though ATC speech should follow the same phraseology, the data from different airports substantially differ due to local conventions, speakers accent and rate of speech. All this together creates a considerable domain shift. Current ASR engines on the ATC domain are tailored to a particular airport.[9] Our ambition was, however, to collect and release annotated and pseudo-labeled recordings from many airports. This fosters the training of more airport-agnostic ASR, NLU, and SLU systems. Previous work has demonstrated that non-ATC audio datasets like LibriSpeech[10] (Panayotov et al., 2015) do not match the ATC acoustics and its use does not help in the ASR training (Zuluaga-Gomez et al., 2020a).

4. Applicability on general spoken language understanding. Even though the *ATCO2 corpus* aims at a niche application (air traffic control communications), we believe that general-purpose research on NLU/SLU can widely benefit from this corpus. Most of the current benchmarks on SLU are widely saturated, where the performances (e.g., F1-scores) are near perfect, a couple of examples are ATIS (Hemphill et al., 1990) or SNIPS (Coucke et al., 2018)

---

9. Other EU-funded projects, like MALORCA or HAAWAII, only focus on developing ASR tools for one or at most two airports per project.

10. This also includes other popular corpora, such as, CommonVoice (Ardila et al., 2019) or Switchboard (Godfrey et al., 1992).

datasets. Differently, *ATCO2 corpus* is composed of very noisy voice recordings (often below 15 dB SNR). The audio data is collected from devices (see Figure 4) owned by a community of volunteers (see §4.1), thus, it is more natural to find noisy data. This, in turn, increases the challenge of standard ASR systems, e.g., word error rates (WERs) of ∼30% or above; see our previous baselines in Zuluaga-Gomez et al. (2023c); Kocour et al. (2021b). We hope that the research community will build upon the *ATCO2 corpus*, fostering the research in the fields of ASR and NLP for ATC communications.

The paper is organized as follows: Section 2 covers previous work on ASR and NLP directed to ATC communications. Section 3 explains our proposed methodologies for the standardization of ATC communications transcription process. The data collection protocol, pre- and post-processing steps undertaken during the transcription process of *ATCO2 corpora* are described in Section 4. *ATCO2 corpora* data statics are reviewed in Section 5. Section 6 and 7 convey the proposed baselines on ASR and NLP, respectively. Section 8 covers the main legal and ethical implications of ATC data collection. Finally, we conclude this paper in Section 9 with final remarks and prospect of future work.

## 2 Previous Works on Collecting ATC Corpora

Currently, there is a vast diversity of databases related to speech and text tasks that have been promoting advances in artificial intelligence (AI). However, ATC communications are still considered an under resourced and underexplored area (Zuluaga-Gomez et al., 2023c,d). Despite the growing interest in text and speech technologies for ATC, there is no commercial ASR engine due to: (i) deficiency in terms of required performance (under 5% WER Ohneiser et al. (2021)), and (ii) lack of large-scale annotated speech data. The cost of data collection and transcription is impractical, even more when transferred to a new airport, which requires an additional round of collection and labeling.

### 2.1 Background

Research attempting to aid ATCos by ASR date as back as late 70s'. Initial, systems aimed at isolated word recognition, speaker verification and commands recognition for military applications (Beek et al., 1977). Exploratory research towards integration of ASR technologies to aid ATCos started in the late 80s with Hamel et al. (1989). Several other research directions target user-friendly and robust automatic systems to train ATCos, or the so called 'pseudo-pilots' (Matrouf et al., 1990). Akin training systems have been proposed by Tarakan et al. (2008); Ferreiros et al. (2012); Zhang et al. (2022); Lin et al. (2021a,b).

We shortlist the three biggest European-based projects that aim at developing speech and text-based tools to aid ATCos in their daily tasks. Initially, MALORCA project[11] was a step forward in demonstrating that ASR tools can cut down ATCos workload (Helmke et al., 2016) while increasing the overall efficiency (Helmke et al., 2017). Then, HAAWAII project[12] has led initiatives to extract key entities (e.g., NER or SF) in the transcribed dialogues produced by an ASR system (Kleinert et al., 2021). Finally, ATCO2 project (our

---

11. MAchine Learning Of speech Recognition models for Controller Assistance, website: `http://www.malorca-project.de/wp/`.

12. Highly Automated Air traffic controller Workstations with Artificial Intelligence Integration, website: `https://www.haawaii.de/wp/`.

corpora) aimed at reducing the human work needed to develop ASR and SLU tools for ATC, mainly by integrating semi-supervised techniques to improve the pseudo-transcription process (Kocour et al., 2021b; Zuluaga-Gomez et al., 2020b).

In addition, *ATCO2* corpora has been used for developing virtual simulation pilots in (Prasad et al., 2022b) which was later extended in (Zuluaga-Gomez et al., 2023b).The virtual simulation pilot system receives spoken ATC audio from ATCo trainees, and it performs ASR and understanding to later provide a pilot's read back.

While the MALORCA and HAAWAII corpora are not public, in ATCO2 we developed a pipeline to collect large quantities of ATC speech data, which are distributed to the public through ELDA. Finally, note that ATCO2 project data has been briefly introduced and used for development of ASR and NLP solutions. However, this paper conveys the details of the ATCO2 platform. We also explore multiple speech and NLP tasks, namely: ASR, named-entity recognition, callsign recognition and speaker role detection.

## 2.2 Command-related ATC Corpora vs Standard Corpora

The ATC speech corpora differ vastly from the standard ASR-training corpora. The root of the discrepancy goes from grammar and vocabulary to audio quality. The standard corpora like Librispeech (Panayotov et al., 2015), Common Voice (Ardila et al., 2019), AMI (McCowan et al., 2005) or TED-LIUM (Rousseau et al., 2012) either target conversational, read or spontaneous speech while also being mostly regarded as 'clean speech'. In contrast, ATC speech is heavily accented, comprises considerably higher noise levels (e.g., below 15dB signal-to-noise (SNR) ratio), high speech rate and artifacts. Previous work has demonstrated that the use of standard corpora do not bring significant improvement in ASR for ATC (Zuluaga-Gomez et al., 2020b).

Even though, ATC English corpora share common vocabularies, there is still a domain shift caused by non-native speakers. One example are ATCos from Switzerland. Even though they are from the same country, accent varies depending on the location. This, in turn, increases the challenge of developing robust enough systems that generalize well across different in-domain environments. Therefore, a non-adapted ASR or NLP system will provide significantly worse performance due to unseen accents, out-of-vocabulary (OOV) words or simply due to discrepancy in the recording procedure.

Further details about the corpora produced by previous projects related to ATC communications are covered in Table 1. Current ATC corpora can be classified into (i) public and (ii) private databases. Public databases normally require a small fee and sometimes are restricted to only-research purposes, such as the multilingual corpus, ATCSpeech (Yang et al., 2020; Lin et al., 2021b). While private corpora[13] are only usable along the concerned project, for instance, to train and test their ATC-related systems. One example is MALORCA, where the two produced corpora, *Prague* and *Vienna* datasets, are widely used only by partners from HAAWAII and ATCO2 projects.

---

13. In fact, nearly all ongoing and former projects in the area of ATC prohibit the release of databases, code, and AI models due to privacy issues.

Table 1: Air traffic control communications related databases. This table list public and private ATC databases. The *ATCO2 corpora* are public databases. †full database after silence removal. ††speaker accents depend on the airport's location, however, the accent of pilots are not known at any time of the communication due to privacy regulations.¶open-sourced at `https://github.com/sculyi/ASR-Corpus/tree/master`.

| Database | Details | Licensed | Accents | Hours† | Ref |
|---|---|---|---|---|---|
| *Private databases* | | | | | |
| HAAWAII | Real data from Iceland and London airports | ✗ | Icelandic, British | 47 | Zuluaga-Gomez et al. (2023c) |
| MALORCA | Real data: LOWW and LKPR | ✗ | German, Czech | 13 | Kleinert et al. (2018) Srinivasamurthy et al. (2017) |
| AIRBUS | Real data from LFBO | ✗ | French | 100 | Delpech et al. (2018) |
| VOCALISE | Real data from terminal maneuvering area and area control center in France | ✗ | French | 150 | Graglia et al. (2005) |
| ENAC | Real data from two French enroute control centers and one major airport | ✗ | French | 22 | Lopez et al. (2013) |
| *Public databases* | | | | | |
| ATCOSIM | Simulated in studio, added cockpit noise. Recordings split by gender (Male/Female) | ✓ | Swiss German, German, French | 10.7 | Hofbauer et al. (2008) |
| UWB-ATCC | Real data from LKPR | ✓ | Czech | 13.2 | Šmídl et al. (2019b) |
| LDC-ATCC | Real data from 3 US airports: KBOS, KDCA and KDFW | ✓ | American English | 26.2 | Godfrey (1994) |
| HIWIRE | Simulated in studio, ATC prompts, added cockpit noise | ✓ | French, Greek, Italian, Spanish | 28.7 | Segura et al. (2007) |
| ATCSpeech¶ | Real accented Mandarin Chinese and English | ✓ | Chinese and English | 57.8 | Yang et al. (2020); Lin et al. (2021b) |
| *Released corpora by ATCO2 project* | | | | | |
| ***ATCO2 corpora*** | Data from different airports and countries | | | | |
| *ATCO2-test-set* | Transcribed audio | ✓ | Multiple†† | 4 | Kocour et al. (2021b) |
| *ATCO2-PL-set* | Pseudo-transcribed audio | ✓ | Multiple†† | 5281 | |
| *Free access databases releseased by ATCO2 project* | | | | | |
| *ATCO2-test-set-1h* | 'ASR dataset': `https://www.atco2.org/data` | ✓ | Multiple†† | 1 | Kocour et al. (2021b) |
| *ATCO2-ELD set* | 'LID dataset': `https://www.atco2.org/data` | ✓ | Multiple†† | 26.5 | Szöke et al. (2021) |

## 3 How To Transcribe Air Traffic Control Audio Data?

This section reviews our collective efforts to provide an unambiguous and clear protocol on how to annotate ATC speech data. We aim at avoiding as much as possible the errors caused by OOV words and phonetic dissimilarities (e.g., "hold in position" and "holding position", or, "climb to two thousand" and "climb two two thousand"). We also rely on the International Civil Aviation Organization (ICAO), which defines a standard phraseology (ICAO, 2020) to

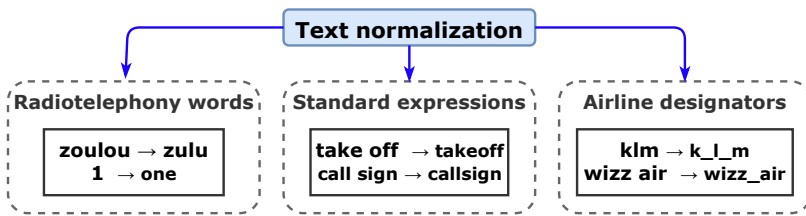

Figure 3: Text normalization applied to the transcription process of *ATCO2* corpora. Further examples in Appendix B.

reduce these errors during the communications.[14] This section first formulates an approach to unify transcripts from different public ATC databases (see Table 1 and Appendix B). Second, it discusses how to craft a vocabulary and lexicon tailored to ATC communications.

### 3.1 Unification of Transcripts

Differently from other corpora, ATC is regulated by a set of rules and a defined grammar. We have seen that in the already available databases (see Table 1), the transcription format and its rules widely diverge. There is not a clear path to follow when it comes to data collection and transcription. Even for a single database, it can be challenging to specify and follow their transcription conventions. One example are numbers. In ATC communications, numbers are key for addressing the aircraft or obtaining its speed or altitude. Several databases have opted to annotate numbers as digits (e.g., $1 \rightarrow 1$), while other databases have chosen to use words (e.g., $1 \rightarrow one$). That is why the *ATCO2* corpus also aims at providing a set of good practices and rules to correctly and unambiguously annotate ATC dialogues. Therefore, if we succeed in reducing the variability of "writing the same thing in many ways", we can considerably reduce the errors and ambiguity committed by subsequent systems in the pipeline, e.g., ASR or NLU. The next logic step, before starting the transcription process, is to define a set of rules to either, unify the transcripts of already available corpora[15] or, to annotate a new corpus.

In general, we apply three different text normalization approaches, as shown in Figure 3. This aims at fostering good practices while labeling the ATC audio. The mapping rules are applied as text filters. We use them to reformat the human-created gold transcripts for the ASR and NLP systems. For the transcription of *ATCO2-test-set corpus*, we also defined a transcription manual that is reachable from the transcription platform `https://www.spokendata.com/atco2`. Finally, we redirect the reader to Appendix B, where additional mapping rules are covered in Table 9.

### 3.2 Lexicon

The lexicon is a table that maps words into pronunciations (phoneme-strings). It is a resource used by the HMM-based ASR tools. Our lexicon is based on the CMU Pronouncing

---

14. Previous work in (Helmke et al., 2018) proposes a novel ontology agreed by several European institutions to annotate unambiguously ATC dialogues.

15. We use these rules to normalize the transcripts of *UWB-ATCC* and *LDC-ATCC* databases (see Table 1 and 3) for experimentation.

Table 2: Samples taken from *ATCO2-test-set corpus* and the Prague Airport (LKPR) subset ID: LKPR_RUZYNE_Radar_120_520MHz. Each callsign (named-entity) is highlighted in blue. IDs are assigned during the data preparation. See the GitHub repository.

| IDs | Transcripts |
|---|---|
| 20201025-091112-A | oscar kilo papa mike bravo descend flight level one hundred |
| 20201025-140929-A | eurowings seven alfa bravo turn right heading two one zero cleared i_l_s approach runway two four report established |
| 20201025-174652-A | lufthansa eight hotel romeo praha radar good evening identified |
| 20201026-142311-A | ryanair nine two bravo quebec turn right heading zero nine zero |

Dictionary,[16] which defines the phoneme set, and it is used as the training data for the grapheme-to-phoneme (G2P) module that synthesizes pronunciations of "new words". We gather all possible words from the training corpora, and we add some other words from different resources. We synthesize the pronunciations by G2P model trained with the Phonetisaurus[17] tool.

The 'spelled acronyms' like "KLM" (pronounced as *"k ey eh l eh m"*) are treated separately and represented as a single token (e.g.,'k_l_m'). We also add manually created pronunciations for some non-English words that cannot be guessed by the G2P model. All the 'word tokens' in the lexicon are in lower-case. We keep only words relevant to ATC domain, i.e., words present in ATC transcripts or other resource. The lexicon contains 29k unique word-symbols.

Our strategy to mitigate the out-of-vocabulary problem is based on enriching the lexicon as much as possible as part of the data preparation. We enriched the lexicon with a list of airline designators for callsigns (partly manually updated).[18] Also, we added all five-letter waypoint[19] names in Europe retrieved from open-source project Traffic.[20] Finally, we introduced some additional words, such as countries, cities, airport names, airplane models and brands, ATC acronyms, etc.

Table 2 shortlist some examples taken from the *ATCO2-test-set corpus* after applying all the transcript normalization steps.

## 4 Data Collection

This section describes the collection and pre-processing of the audio data and ATC metadata in the *ATCO2* corpus. An overview of the data processing pipeline is given in Figure 2. First, the data is collected via very-high frequency (VHF) radio receivers that are owned by a community of volunteers. Then, the audio and metadata are uploaded to OpenSky Network

---

16. Dictionary at: `http://www.speech.cs.cmu.edu/cgi-bin/cmudict`.

17. GitHub repository: `https://github.com/AdolfVonKleist/Phonetisaurus`.

18. List taken from Wikipedia: `https://en.wikipedia.org/wiki/List_of_airline_codes`.

19. A waypoint is an intermediate point or place on a route or line of travel, a stopping point or point at which an aircraft's course is changed.

20. *Traffic* project: `https://pypi.org/project/traffic/`.

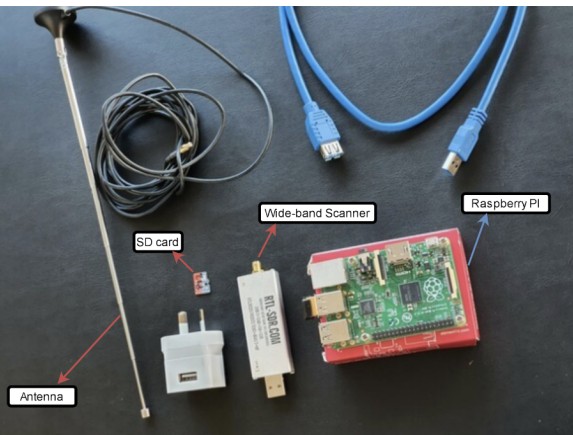

Figure 4: A set of items needed to set up a VHF receiver.

(OSN) servers[21] via Internet. Finally, the collected data is processed on a ReplayWell server[22] via REST API, and part of it is selected for human transcription (see Appendix A). The ReplayWell server hosts a major part of the data processing pipeline. We also rely on a community of volunteers for transcription.

## 4.1 Data Feeders

The data feeders are volunteers who capture ATC voice and upload it to OSN servers. The Data Feeders are typically aviation enthusiasts with possible prior operational experience, or people with an interest in aviation technologies (e.g., people doing domain related research, radio amateurs, etc.). To become a feeder, one needs to own a VHF receiver, which consists of an antenna, software defined radio (SDR) and a computer connected to the Internet. Affordable and popular options such as an RTL-SDR dongle and Raspberry Pi single board computer work sufficiently well in most cases. Quality of recorded ATC data varies depending on the equipment utilized during its collection (properly tuned gain parameter, position of antenna, DSP processor in the radio receiver). As an example, an affordable setup can be built with a *Sirio Md 118-137* antenna and an *RTL-SDR* radio receiver dongle (RTL2832U with 8-bit analog-to-digital converter), this setup is similar to items in Figure 4. For better quality, we recorded with a *Watson WBA-20* antenna and a *SDRPlay - RSP1A* radio receiver, which has a 14-bit analog-to-digital converter.

In some countries, it might be prohibited by law to record air traffic management (ATM) related data. The data feeders should check the applicable regulations before recording and feeding the data to the Internet. If you are interested in becoming a data feeder, please follow the instructions in the 'feeder zone' website: `https://ui.atc.opensky-network.org/set-up`.

---

21. OpenSky Network is a non-profit association based in Switzerland. It aims at improving the security, reliability, and efficiency of the airspace usage by providing open access of real-world air traffic control data to the public. The OpenSky Network consists of a multitude of sensors connected to the Internet by volunteers, industrial supporters, and academic/governmental organizations.

22. The whole pipeline runs live in the following URL: `https://www.spokendata.com/atco2`.

### 4.2 Data Annotators

The annotators are people who produce transcripts of ATC voice communications. These transcriptions also include assigning speakers roles and tagging of named entities. During the ATCO2 project, we relied on both the volunteers and paid transcribers. Volunteers with knowledge of ATC phraseology are ideal, but not strictly required.

Currently, we use our data processing pipeline (see Figure 2), which generates the initial transcripts and NLP tags. Pre-transcribing with AI tools speeds-up the overall transcription process. If you are interested in becoming an annotator, please create an account in the SpokenData transcription platform: `http://www.spokendata.com/atco2`. All the transcribed data within ATCO2 project's life was packaged and released as the *ATCO2-test-set corpus*. Both the data feeders and annotators will have access to the data they provided.[23]

### 4.3 Data Processing Pipeline

The steps from ATCO2's automatic *data processing pipeline* in Figure 2 are briefly described below:

**Segmentation and demodulation:** the RTL-SDR radio receiver tuned to a frequency provides a data stream in IQ format. The RTL-SDR software [24] has an in-built Voice Activity Detection (VAD) segmentation features. This is based on detecting abrupt changes of energy in the signal. The IQ signal is demodulated into a wave file with *csdr* software. [25] The *csdr* software is configured to remove DC offset, and we don't use automatic gain control. It is important to tune the *gain* parameter in the RTL-SDR software, so the audio is both well audible and not overboosted.

**Segment-based gain control:** the signal from distant airplanes can be weak. We noticed speaker turns are often separated by spikes in the waveform. The spikes arise from the window-based DC-offset removal in *csdr*. We detect these spikes and increase volume in segments separated by the spikes when needed.

**Signal-to-noise ratio filtering:** the next processing step is "signal-to-noise ratio filtering". We discard recordings that are too noisy, but audio files with moderate level of noise are not discarded. We use WADA-SNR (Waveform Amplitude Distribution Analysis) (Kim and Stern, 2008) to estimate the SNR. WADA-SNR is based on analysis of shape of distribution over samples in a speech waveform. The non-speech parts are removed by a speech activity detection (SAD) tool (Plchot et al., 2018) with a 'tight' preset, leaving almost no non-speech parts marked as speech.

**Acoustic-based speaker diarization**: a single recording can have multiple speakers in it, so the per-speaker segments are identified by diarization. We do it before the automatic transcripts are generated, so the NLU modules always process segments of a single speaker. Also, the annotators are asked to eventually rectify the per-speaker segments.

For details of the acoustic diarization VBx model, the reader is referred to (Landini et al., 2022). This model uses a Bayesian hidden Markov model (BHMM) to find speaker clusters in a sequence of x-vectors. The x-vector extractor uses DNN architecture based on

---

23. More information on the official Opensky Network website: `https://opensky-network.org`.
24. RTL-SDR radio receiver software: `https://github.com/szpajder/RTLSDR-Airband.git`
25. CSDR sofware defined radio: `https://github.com/ha7ilm/csdr`

Table 3: Train and test sets configuration for baseline experiments. [†]entire *ATCO2-PL corpus* used for training our ASR modules, see Table 1. [¶]this subset filters out recordings that do not contain speaker role tags (used for speaker role detection). However, we report results on the full *ATCO2-test-set corpus* for the ASR experiments.

| | Statistics | | | |
|---|---|---|---|---|
| **Dataset** | Nb. samples [k] | Duration [h] | SNR [dB] | Public |
| *ATCO2-PL-set (train)*[†] | 3072 | 5281 | any | ✓ |
| *ATCO2-test-set (test)*[¶] | 3 | 3.4 | ≤15 | ✓ |

`ResNet101` (Landini et al., 2022). In the first step, Agglomerative Hierarchical Clustering (AHC) is applied to the extracted x-vectors. Then, Variational Bayes HMM over x-vectors is applied using the AHC output.

**Automatic speech recognition:** our ASR system has been trained on several publicly available databases (Šmídl et al., 2019b; Godfrey, 1994; Šmídl et al., 2019a; Hofbauer et al., 2008) and some private databases (AIRBUS, MALORCA). It is a hybrid ASR system trained with the Kaldi (Povey et al., 2011). The system is covered in more details in §6, and also in (Kocour et al., 2021b). The ASR output is a confusion network. It is a 'sausage-like' structure with lists of alternate words in bins, and word-confidences in each bin sum up to one. This task is covered in this paper.

**English language detection:** we deployed an *English language detection* system (ELD) to separate non-English utterances from the input stream of data. Specifically, we used an NLP-based system that processes ASR output transcripts with word confidences. This system was more robust and better than standard acoustic-based ELD system (Szöke et al., 2021). Another benefit from using an NLP system is that it can jointly use logits or probabilities outputs from different ASR systems, which further can boost the results. Our ELD tool was previously covered in (Kocour et al., 2021b; Szöke et al., 2021).

**Post-processing by NLP:** in ATCO2 project, we focused on extracting knowledge from the text produced by the ASR system. *ATCO2-test-set corpus* contains rich metadata extracted with different NLP and NLU based modules. Specifically, we performed three tasks:

- *Callsign recognition*: locate the callsign and convert it to code, such as `"KLM91G"`

- *ATCo/pilot classification*: decide who is speaking in the entire utterance

- *ATC-Entity recognition*: highlight the callsign, command and value entities in text

Further details about our NLP/NLU modules are covered in (Kocour et al., 2021b). Information about integration and pre-processing pipeline is in Appendix A. This task is covered in this paper.

## 5 Datasets

Here, we describe in details the datasets for our baseline experiments. We evaluate on the *ATCO2-test-set corpus* as an in-domain test set, and *MALORCA-Vienna test set* as an unseen

Table 4: Stats about the collected databases per airport. Duration, SNR, language scores and contextual data columns report the mean and standard deviation (mean/**std**) per sample. Each recording/sample contains one or more segments (we provide timing information in RTTM format). [†]abbreviation in IETF format. [††]total number of segments and accumulated duration of speech hours (after voice activity detection) per airport.

| Database | | Metadata | | | | Contextual data | |
|---|---|---|---|---|---|---|---|
| ICAO - Airport | Accent[†] | # Segments[††] | Dur. [sec] | SNR [dB] | Lang Score | # n-grams | # entities |
| *English Data (language score $\geq 0.5$)* | | | | | | | |
| EETN - Tallinn | et | 79 k/**131 hr** | 6.0/**3.4** | 4.6/**7.8** | 0.96/**0.08** | 104/**26** | 37/**9** |
| EPLB - Lublin | pl | <1 k/<**1 hr** | 13.3/**8.0** | 2.5/**8.2** | 0.94/**0.11** | 19/**10** | 4/**2** |
| LKPR - Prague | cs | 999 k/**1762 hr** | 6.4/**4.3** | 14.2/**8.2** | 0.95/**0.09** | 230/**95** | 70/**30** |
| LKTB - Brno | cs | 401 k/**888 hr** | 8.0/**14.4** | 4.1/**15.7** | 0.88/**0.15** | 49/**35** | 15/**10** |
| LSGS - Sion | fr-ch | 168 k/**330 hr** | 7.1/**4.8** | 10.0/**8.0** | 0.87/**0.15** | 56/**23** | 20/**8** |
| LSZB - Bern | gsw | 324 k/**699 hr** | 7.8/**5.0** | 15.4/**10.7** | 0.9/**0.13** | 101/**42** | 36/**15** |
| LSZH - Zurich | gsw | 470 k/**921 hr** | 7.0/**4.6** | 7.8/**7.7** | 0.94/**0.1** | 526/**179** | 169/**55** |
| LZIB - Bratislava | sk | 9 k/**24 hr** | 8.8/**6.9** | 5.4/**8.7** | 0.86/**0.15** | 68/**27** | 22/**8** |
| YBBN - Brisbane | en-au | 105 k/**170 hr** | 5.8/**4.1** | 10.2/**5.8** | 0.93/**0.1** | 268/**86** | 95/**30** |
| YSSY - Sydney | en-au | 49 k/**77 hr** | 5.7/**9.2** | 3.1/**7.0** | 0.92/**0.11** | 495/**148** | 174/**52** |
| others - others | others | <1 k/<**1 hr** | 5.0/**6.7** | 4.0/**7.4** | 0.92/**0.11** | 55/**260** | 16/**78** |
| *Non-English Data (language score $< 0.5$)* | | | | | | | |
| EETN - Tallinn | et | 2 k/**2 hr** | 4.0/**2.4** | 2.9/**10.8** | 0.3/**0.14** | 95/**30** | 33/**11** |
| EPLB - Lublin | pl | <1 k/<**1 hr** | 13.1/**2.7** | -8.4/**12.8** | 0.2/**0.13** | 17/**7** | 4/**1** |
| LKPR - Prague | cs | 105 k/**187 hr** | 6.4/**5.4** | 13.8/**9.3** | 0.18/**0.16** | 217/**97** | 67/**30** |
| LKTB - Brno | cs | 214 k/**611 hr** | 10.3/**19.2** | 6.5/**11.8** | 0.15/**0.15** | 56/**33** | 18/**10** |
| LSGS - Sion | fr-ch | 57 k/**83 hr** | 5.3/**3.6** | 9.8/**9.3** | 0.27/**0.14** | 56/**25** | 20/**8** |
| LSZB - Bern | gsw | 42 k/**55 hr** | 4.7/**3.4** | 13.6/**13.9** | 0.3/**0.13** | 102/**45** | 37/**16** |
| LSZH - Zurich | gsw | 36 k/**49 hr** | 5.0/**4.1** | 2.0/**12.7** | 0.25/**0.15** | 485/**180** | 157/**56** |
| LZIB - Bratislava | sk | 10 k/**26 hr** | 9.0/**7.6** | 7.1/**7.0** | 0.18/**0.15** | 72/**26** | 23/**8** |
| YBBN - Brisbane | en-au | 7 k/**10 hr** | 4.9/**4.8** | 5.9/**12.3** | 0.24/**0.16** | 268/**79** | 95/**28** |
| YSSY - Sydney | en-au | 3 k/**3 hr** | 3.9/**2.3** | 2.7/**10.5** | 0.33/**0.13** | 481/**151** | 169/**53** |
| others - others | others | <1 k/<**1 hr** | 5.2/**6.4** | 3.7/**9.0** | 0.26/**0.15** | 0/**0** | 0/**0** |

airport. The baseline systems are trained purely with the *ATCO2-PL-set corpus* and its automatic transcripts (i.e., pseudo-labels).

## 5.1 ATCO2 datasets

**ATCO2-test-set corpus:** this dataset was built for development and evaluation of ASR and NLP technologies for English ATC communications. The dataset consists of English coming from LKTB, LKPR, LZIB, LSGS, LSZH, LSZB and YSSY airports. We provide two partitions of the data, the *ATCO2-test-set-1h corpus* and the *ATCO2-test-set corpus*. The first corpus contains 1 hour sub-set with open-sourced transcriptions. The latter adds around 3 hours of annotated data, accounting for a total of 4 hours. The amount of data per airport are summarized in Table 5. The recordings of both corpora are mono-channel sampled at 16 kHz and 16-bit PCM, including transcripts, NLP tags and speaker role tags. An example of the XML format is in Appendix C.

**ATCO2-PL-set corpus:** ATCO2 project recorded a large database of ATC voice communications. Altogether, we collected over 5'281 hours of ATC speech from different airports

Table 5: *ATCO2-test-set corpora* split by airports.

| | *ATCO2-test-set* | | *ATCO2-test-set-1h* | |
|---|---|---|---|---|
| **ICAO - Airport** | sentences | words | sentences | words |
| LKPR Prague | 207 | 2686 | 102 | 1254 |
| LKTB Brno | 60 | 854 | 32 | 451 |
| LSGS Sion | 932 | 10183 | 256 | 2684 |
| LSZB Bern | 452 | 5908 | 172 | 2323 |
| LSZH Zurich | 640 | 8123 | 126 | 1764 |
| LZIB Stefanik | 165 | 2256 | 79 | 1051 |
| YSSY Sydney | 1065 | 10434 | 102 | 1058 |
| Sum | 3521 | 40444 | 689 | 10585 |

around the world (see Table 4). In total, we cover ten airports. Table 4 depicts all this metadata per airport, further split by English language score. To the best of the author's knowledge, this is the largest and richest[26] dataset in the area of ATC ever created that is accessible to the public. The automatic transcripts are stored as confusion network, stored as a *CTM* text format extended to have more words per line (confusion network bin):

```
<wav-id> <speaker> <t_begin> <dur> <word1> <conf1> <word2> <conf2> ...
LKPR_Tower_134_560MHz_20211223_154543 A 1.25 0.10 the 0.845 papa 0.042 ...
```

Another view of the data is in Figure 5, where we don't split by the English detection score. The distributions are from the full, 5281 hours dataset. In sub-figure 5a, we see that the majority of data was recorded in Prague, Brno, Zurich, Bern, and Sion airports. This is because we started recording these airport's data early in the ATCO2 project. Next, we also have some data from Brisbane, Talinn, Sydney, Bratislava (STEFANIK) and Lublin. In sub-figure 5b, we see that majority of our data has high English scores (the dashed line 'ALL DATA'). There are some airports that have more non-English utterances: Brno, Bratislava (STEFANIK). And there are some airports, for which the distribution is more uniform than for others: Sion. We know for sure that local language is present at higher quantities in the data from Brno, Bratislava, and Sion airports. And bigger airports usually have a policy to speak only in English, which explains the low number of detections of non-English speech there.

From sub-figure 5c we see that levels of noise differ across the airports. The cleanest signal is for Prague and Bern, while high levels of noise are from Sydney and Lublin, and some noise also for Tallinn and Bratislava. This indicates that the recording setup could be improved. In addition, from sub-figure 5c we can see that the mean SNR levels for ATCO2 is below 20 dB in all cases. This confirms that ATC data is more challenging with regard to audio quality when compared to non-ATC corpora.

Finally, in sub-figure 5d are the distributions of confidences of the automatic transcripts, assigned by the seed ASR system. Majority of the probability mass is in interval $(0.8, 1.0)$ (dashed curve 'ALL DATA'). The highest confidence is assigned to Prague data (highest

---

26. By richest, we mean quality of transcriptions (including pseudo-transcriptions) and amount of metadata per sample. Also, this is the first public database in the area of ATC that contains parallel data: audio, transcripts and labels for NLU tasks.

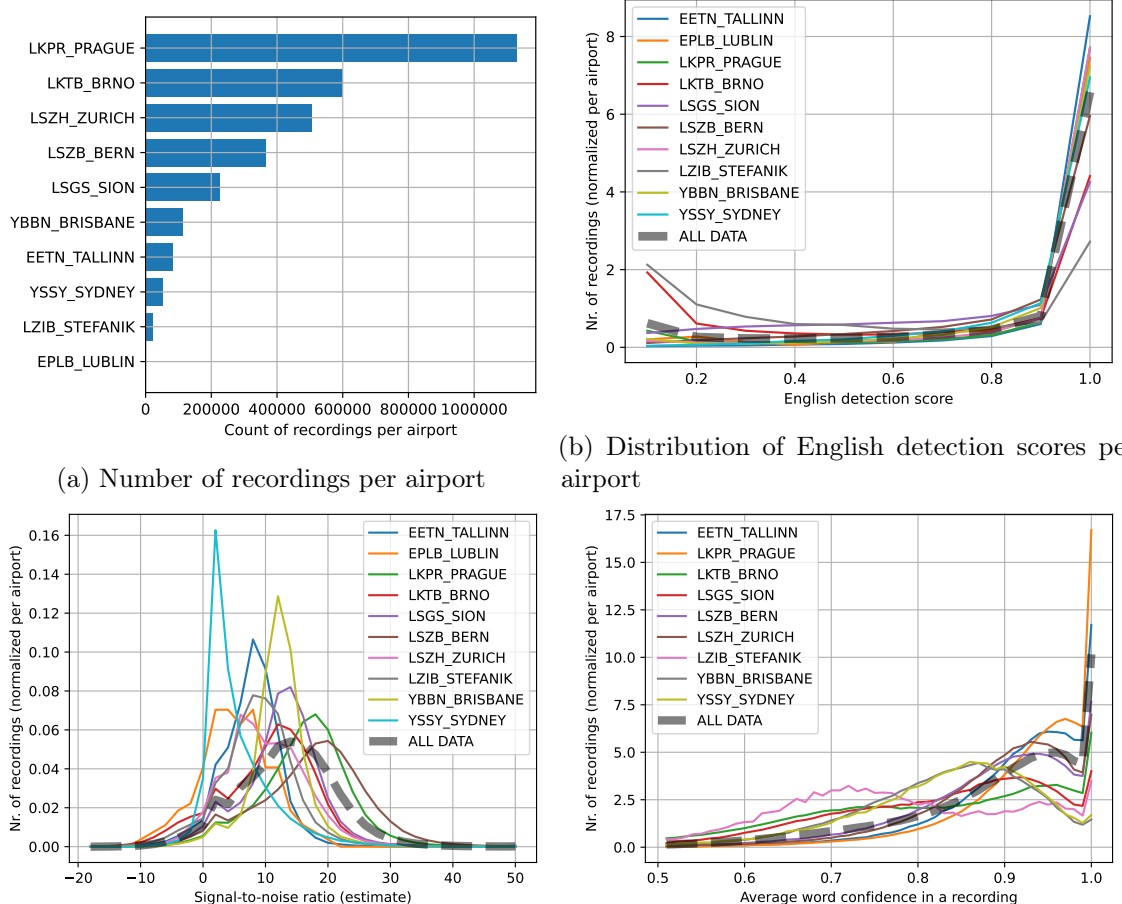

(a) Number of recordings per airport

(b) Distribution of English detection scores per airport

(c) Distribution of Signal-to-noise ratio per airport

(d) Distribution of Confidence scores per airport

Figure 5: Distribution plots of metadata for *ATCO2-PL set corpus*.

peak on right). The lowest confidence have the data from Bratislava, Brisbane, and Sydney airports (distributions with leftmost modes). The higher tails with lower confidences are very likely caused by the non-English speech and noisy signal in the data.

### 5.2 Private Databases

**MALORCA Vienna test set:** The MALORCA Vienna test set is used in baseline ASR experiments as an unseen airport. No Vienna data are in the *ATCO2-PL-set* that use for training the acoustic model and language model. On the other hand, MALORCA Vienna data were present in the training of the seed system for generating the automatic transcripts. So it both unseen and indirectly seen at the same time. The set consists of ATCo speech only, which normally has lower error rates than the pilot speech (Delpech et al., 2018; Pellegrini et al., 2018). The total amount of speech after VAD is 1.9 hours. The audio data is mono-channel sampled at 8 kHz and 16-bit PCM.

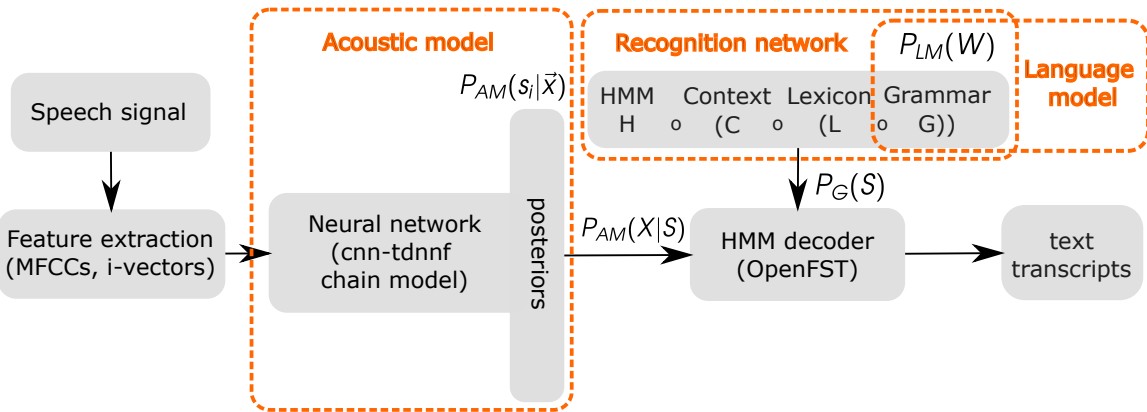

Figure 6: Hybrid-based ASR system. The inference pipeline consists of *feature extraction*, *acoustic matching* by acoustic model and *search* by HMM decoder that uses HCLG recognition network. On the output are text transcripts. Alternatively, the output can be a *lattice* (a graph with alternative decoded paths) or *confusion network* (time-sequence of bins with word-lists having scores).

## 6 Automatic Speech Recognition

ASR systems have an audio signal as its input and produces text transcripts as its output.

At first, we trained a 'seed ASR system' from several existing ATC databases. The seed ASR system is part of the 'data processing pipeline' (see §4.3). This ASR produced the automatic transcripts for the *ATCO2-PL-set corpus* and also the initial transcripts for the *ATCO2-test-set corpus* that were manually corrected. During the ATCO2 project, we worked mainly with hybrid-based ASR systems, trained with the open-source toolkit Kaldi (Povey et al., 2011).

In hybrid speech recognizers, Hidden Markov models support various speech rates, by allowing to "stay in a state" for some time via self-loop transitions. The acoustic scores come from a neural network, and the decoding process is based on searching in the matrix of acoustic scores for state-sequences with plausible transcriptions generated from a pre-compiled recognition network HCLG (i.e., a large HMM graph/model). Thus, HMMs provide a structure for mapping a temporal sequence of acoustic scores into a sequence of states (Morgan et al., 1993; Bourlard and Morgan, 1993), from which the recognized words are extracted.

### 6.1 Inference in hybrid-based ASR

Inference in a hybrid speech recognizer has three stages: feature extraction, acoustic matching and decoding, the overall scheme is in Figure 6.

Feature extraction compresses the waveform into a sequence of fixed-length vectors of low dimension, in our recipe we use high-resolution MFCCs with i-vectors (Saon et al., 2013) appended.

Matching of acoustic units by acoustic model. Here, the acoustic model converts the input features into posterior probabilities of a closed set of acoustic units (phoneme states), whose

time series forms the acoustic score matrix. In our recipe, we use 'chain' model neural network (NN) trained by Lattice-free MMI (Povey et al., 2016), also known as LF-MMI. The NN topology is a 'CNN-TDNNF' architecture with 6 `conv-relu-batchnorm-layer` components followed by 9 `tdnnf-layer` components (Povey et al., 2018), and 2 softmax layers with 2000 outputs each. The acoustic model consists of 12.9 million trainable parameters.

Decoding searches for the most likely word sequence $\hat{W}$ (transcription), in the matrix of acoustic scores. The search explores HMM paths that exist in a recognition network, termed HCLG graph. The standard decoding algorithm is based on two ideas: *token passing* and *beam search*. The search combines scores from the acoustic model, language model and lexicon, as shown in equation (1) :

$$\hat{W} = \text{wrds} \left( \underset{S}{\text{argmax}} \, P_{AM}(S|\mathbf{X})^{\kappa} \, P_G(S) \right) \ . \tag{1}$$

The acoustic model scores are the chain model posteriors $P_{AM}(S|\mathbf{X})$, where $\mathbf{X}$ is the time-series of input features and $S$ is an HMM state-sequence. The language model and lexicon scores are both represented in the graph score $P_G(S)$ that is present in the HCLG recognition network. $\kappa$ is an empirical scaling constant, for chain models the optimal $\kappa = 1.0$. And the function wrds(.) is reading word-sequence from the state sequence $S$ with the maximal score.

The HCLG graph is a Weighted Finite State Transducer (WFST). The HCLG graph is composed of a language model graph G, pronunciation lexicon graph L, context dependency graph C and phoneme HMM graphs H. The HCLG graph contains graph costs $P_G$ that originate from its source graphs, while the most important source is the language model. This was the description of a hybrid ASR system.

The other paradigm type for ASR development is End2End ASR (Chen et al., 2021; Baevski et al., 2020). These models can be trained with (i) Connectionist Temporal Classification (CTC) loss (Graves et al., 2006; Graves and Jaitly, 2014), (ii) attention-based encoder-decoder modeling (Chorowski et al., 2015) or (iii) hybrid (Watanabe et al., 2017), which uses (i) and (ii). However, End2End systems require more training data to achieve good performance.

Hybrid systems remain one of the best and more flexible approaches for building ASR engines. The HMM-DNNs based ASR are used in the current state-of-the-art systems for ASR in ATC communications (Zuluaga-Gomez et al., 2020b; Srinivasamurthy et al., 2017). Hybrid-based ASR systems train independently the Acoustic model and the Language model. The language model is trained on a text corpus, which allows incorporating text resources without the necessity to have the corresponding audio. The hybrid ASR relies on a word-based lexicon, and words that are not in the lexicon or language model cannot be hypothesized by ASR decoder (out-of-vocabulary word problem – OOVs).

## 6.2 Hybrid-based ASR for air-traffic control communications

We use the same ASR system both for ATCos and pilots. The training recipe and databases for our 'seed ASR system' (including the train sets in Table 1) are covered in (Kocour et al., 2021b; Zuluaga-Gomez et al., 2020a,b, 2021). Briefly, we used AIRBUS, MALORCA Vienna, ATCOSIM, UWB-ATCC, LDC-ATCC, HIWIRE and N4-NATO databases. In total, these form a database of $\approx 135$ hours. We augmented this database with noises captured from

LiveATC. And the data were further augmented by speed perturbation (i.e., 135+700 in Table 6). Due to data license issues with some databases, this ASR system can be only used for research.

In a later stage of the ATCO2 project, we experimented with contextual adaptation and semi-supervised training. We later integrated these technologies into the 'seed ASR system'.

The contextual adaptation improves the accuracy of the ASR system by feeding-in a rapidly changing contextual information. Based on surveillance data, we suggested a list of nearby callsigns into the recognizer (Kocour et al., 2021a). This was done by applying a boosting WFST graph to HCLG or lattice. In HCLG boosting, we give score discounts to individual words, while in Lattice boosting, the score discounts are given to word sequences. In addition, the lattice boosting was used when generating the automatic transcripts for *ATCO2-PL-set corpus.* Moreover, offline (Nigmatulina et al., 2022, 2021) and online (Nigmatulina et al., 2023) lattice and language model boosting have explored with the ATCO2 corpus.

The semi-supervised training was used to improve ASR accuracy by retraining the acoustic model (Kocour et al., 2021b) on a mixture of manually and automatically transcribed data. ATCO2 data with automatic transcripts were mixed with transcribed data from other databases. We used per-frame gradient weighting by word confidences to de-weight data with unreliable transcripts. We further performed experiments in (Zuluaga-Gomez et al., 2021). Here, we applied callsign boosting when generating automatic transcripts for semi-supervised learning, and we obtained 17.5% relative WER improvement measured on the callsign words.

### 6.3 Baseline experiments

During the ATCO2 project, we collected 5281 hours of ATC audio data from several airports. We processed the data with our automatic pipeline (see Figure 2) that filters the data and produces automatic transcripts. Inside the pipeline, there is an ASR system that is described in §6 and also in our previous work (Kocour et al., 2021b).

The purpose of these baseline experiments is to demonstrate what can be achieved with the data we collected and released in ELDA catalog. From these automatic transcripts, we can bootstrap and build a new ASR system, without having licensing problems that exist for some other databases (Table 1). We built a new language model from all the generated transcripts. And, we experimented with training acoustic models on various subsets of the audio data. We re-used the lexicon from the 'seed ASR system'.[27]

The baseline experiments are described in Table 6, where we computed WERs on three test sets: *ATCO2-test-set*, *ATCO2-test-set-1h* and *MALORCA Vienna* test set. Each model is tagged with the number from the first column of Table 6. The MALORCA Vienna test set represents clean ATCo speech from an unseen airport.

#### 6.3.1 Analysis of ASR systems from Table 6

In 1) we built the acoustic model and language model on all the data in the ELDA package, including the data that the English detector identified as non-English. In 2) we excluded the non-English data, and from now on, the Language Model is always trained from transcripts of this 4'500 hours dataset (except for seed system). In 3) we set the filtering thresholds higher to >0.7 ELD (English detection), >0 dB SNR and >0.8 CNET score (average word

---

27. See: `https://github.com/idiap/atco2-corpus/tree/main/lm/lexicon`

confidence in a confusion network of recording). In WER results for 1) 2) 3), we see that the *ATCO2 test sets* results stay similar, while the WER for MALORCA test set improves with stronger data filtering. In 4) we realized that it is not a good idea to discard too much noisy data by filtering >16 SNR. Next, in 5) we randomly selected 3600 hours from the 4500 hours dataset. This was to cross-check with the filtering we previously used in 3). To our surprise, the results are marginally better when randomly selecting the data. Next, in 6) 7) 8) we continued randomly selecting subsets from the 4'500 hours dataset. This degraded the performance of MALORCA Vienna on 7) 8) by up to 2% WER. From the results, we notice that WER for ATCO2 test-sets is nearly constant, except 4). It seems that WER in the automatic transcripts is an important factor. The automatic transcripts are used as training targets, and the WER in transcripts pre-determines the performance of the trained system. The amount of training data is possibly less important, however the generalization to a new airport (MALORCA Vienna) is better with larger volumes of training data of 2'500 or 3'600 hours in ASR systems 5) 6).

### 6.3.2 Baseline ASR system without ATCO2

For completeness, we also add WERs of the seed system 9). The seed system has few percent higher WER for ATCO2 test-sets. For MALORCA Vienna, the seed system works as good as 4.8% WER, as MALORCA Vienna corpus was present in its training data.

### 6.3.3 Baseline ASR system with End2End models

In the experiment 10) of Table 6 we perform an early exploration of End2End ASR modeling with large-scale self-supervised models. We selected the XLSR model Conneau et al. (2020), which has been pre-trained on 56'000 hours of speech data from ∼53 different languages. We fine-tune the seed model with LF-MMI criterion, following the approach in Vyas et al. (2021). For this experiment, we use slightly larger dataset as the one used for model 9). We use fairseq Ott et al. (2019) for fine-tuning, while Kaldi at decoding time. The fine-tuning is run for 14'000 updates steps which amounts to 7 epochs. The initial learning rate is $3e^{-5}$ with tri-stage decay.

### 6.3.4 Future work related to ASR

In this paper, we mostly focus on the data novelty brought by the release of ATCO2 corpora. This includes, data collection, pre-processing, packaging and pseudo-transcriptions. However, there are two aspects with regard to ASR that are left as future work: (i) investigating different End2End architectures for ASR. This includes: full attention-based (Vaswani et al., 2017) systems such as Conformer (Gulati et al., 2020) and BranchFormer (Peng et al., 2022), or more efficient ASR systems such as Conmer (Radfar et al., 2023) or HyperConformer (Mai et al., 2023). (ii) exploration of PyTorch-based toolkits for End2End ASR such as PkWrap (Madikeri et al., 2020), SpeechBrain (Ravanelli et al., 2021), ESPnet (Watanabe et al., 2018), or NeMo toolkits (Kuchaiev et al., 2019).

Table 6: Performance on *ATCO2-test-set corpus*. The ASR system is built from the automatic transcripts of the *ATCO2-PL-set corpus* or from only supervised data (i.e., baseline ASR systems without ATCO2). We use two *ATCO2-test-set corpus* splits, and MALORCA Vienna (Kleinert et al., 2018) as an unseen airport. The data filtering is done according to: ELD (English language detection), SNR (signal-to-noise) ratio, and CNET (average word confidence in the recording). †seed model trained with Kaldi framework. ¶fine-tuned XLSR model (Conneau et al., 2020) with fairseq and decoding with Kaldi framework, see Vyas et al. (2021).

| System | Training hours | Use ATCO2 | WER | | | Data Selection Method | | | |
|---|---|---|---|---|---|---|---|---|---|
| | | | ATCO2 test-set (4h) | ATCO2 test-set 1h | MALORCA Vienna | ELD | SNR | CNET | Note: |
| 1) | 5281 | ✓ | 22.3 | 15.8 | 11.1 | any | any | any | full *ATCO2-PL-set corpus* |
| 2) | 4500 | ✓ | 22.5 | 15.7 | 10.0 | >0.5 | any | any | remove non-English |
| 3) | 3600 | ✓ | 22.5 | 15.8 | 9.3 | >0.7 | >0 | >0.8 | remove low-confidence |
| 4) | 1500 | ✓ | 23.4 | 16.7 | 11.9 | >0.5 | >16 | any | remove low SNR (<16) |
| 5) | 3600 | ✓ | 22.4 | 15.4 | 9.0 | >0.5 | any | any | random from 4500 hr set |
| 6) | 2500 | ✓ | 22.6 | 15.8 | 9.0 | >0.5 | any | any | random from 4500 hr set |
| 7) | 1500 | ✓ | 22.5 | 15.8 | 10.6 | >0.5 | any | any | random from 4500 hr set |
| 8) | 500 | ✓ | 22.5 | 15.7 | 11.0 | >0.5 | any | any | random from 4500 hr set |
| 9) | 135 | ✗ | 26.6 | 18.6 | 4.8 | - | - | - | †CNN-TDNNF |
| 10) | 190 | ✗ | 24.9 | 17.5 | 5.5 | - | - | - | ¶XLSR-53 |

# 7 Natural Language Understanding

Until the previous decade, research on ATC was directed at only transcribing as accurate as possible the dialogues between ATCos and pilots. However, transcription is only one part of the story and further information, such as, entity highlighting (also known as intent and slot filling) or speaker role detection is imperative in real-life ATC control rooms. The process of parsing these high-level entities from ATC audio can be seen as SLU, or from text as NLU.

Previous work has already explored different NLP tasks in the area of ATC. For instance, (Lin, 2021) describes a set of entities and elements that are present in ATC communications that are of special interest, e.g., commands and instructions (Zhang et al., 2022). The authors advise that a real-life system should be composed of an ASR module to obtain the word-level transcripts of the communication. Later, a subsequent system should extract ATC-related key entities and then parse them into a specific grammar. We redirect the reader to (Helmke et al., 2018), which developed an ATC-structured grammar accepted by several European institutes. Furthermore, in (Lin, 2021), the process of extracting key entities from audio is summarized in an entire pipeline composed of three submodules. Namely, speaker role detection, intent classification and, slot filling (analogous to NER but on audio level). They aim at inferring the near-future air traffic dynamics, which can aid ATCos in their daily task. In addition, this system can notice communication errors caused by one of the speakers, also known as hear or read back errors. Some exploratory work addressing NLP and NLU on the framework of HAAWAII and ATCO2 projects (see Table 1) is described in (Prasad et al., 2022a; Zuluaga-Gomez et al., 2023d).

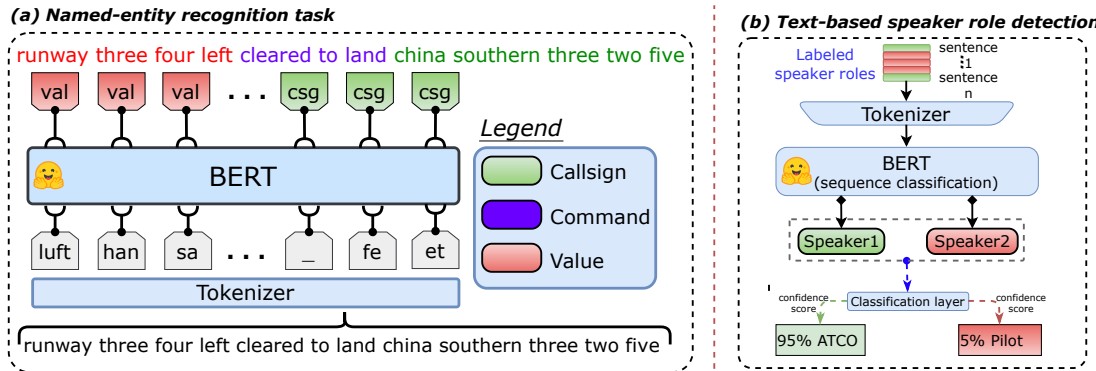

Figure 7: (a) Named entity recognition and (b) speaker role detection based on sequence classification (SC) for ATC utterances. Both systems fine-tune a pre-trained BERT (Devlin et al., 2018) model for ATC tasks. The NER systems recognizes callsign, command and values, while the SC assigns a speaker role to the input sequence.

In this section, we describe our baselines for two tasks related to NLP and NLU.[28] In ATC, in addition to transcripts generated by an ASR system, we can also extract rich metadata from the transcripts and audio. Some examples are, but not limited to:

- ✓ What are the high-level entities in the communication? → named-entity recognition (NER) or slot filling (SF). Previous work in (Nigmatulina et al., 2022) and covered in §7.1,

- ✓ Who is talking? ATCo or pilot → speaker role detection (SRD), sequence classification. Early work on (Prasad et al., 2022a), and covered in §7.3,

- ✗ Is the pilot responding the correct information? → read-back error detection. Previous work in (Helmke et al., 2022) under HAAWAII project, in (Ahrenhold et al., 2023) funded by SESAR 2020 PROSA project (PJ.10-W2), and some others in (Helmke et al., 2021, 2023),

- ✗ Is the communication being uttered in English language? → English language detection (ELD). Previous work in (Szöke et al., 2021).

We present baselines only on the above items marked with ✓, while the items marked with ✗ are, either covered in previous work or left as future research directions. Generally speaking, extracting the above-mentioned information could allow to further fulfill other ATC tasks, e.g., pre-filling radar labels in the ATC control room. Or, for example, decrease the workload of ATCos and increase their efficiency by automating manual and effortful processes. In addition, reducing the overall probability of incidents and accidents due to air traffic management erroneous procedures is a supplementary by-product of introducing AI tools in the ATC control rooms.

---

28. As we work on top of ASR transcripts, these tasks can be also cataloged as spoken language understanding.

### 7.1 Named Entity Recognition

Named entity recognition, or NER, is one of the most explored tasks in the field of information extraction and NLP (Vajjala and Balasubramaniam, 2022). NER aims to locate and classify entities in unstructured text into pre-defined classes or categories. Examples are, persons or organization names, expressions, or, for instance, callsigns or commands in ATC. Initially, NER was based on handcrafted lexicons, ontology, dictionaries, and rules (Grishman and Sundheim, 1996). Even though these systems were interpretable and understandable, they were prone to human errors. Collobert et al. (Collobert et al., 2011) introduced machine learning-based methods for text processing in topics such as part-of-speech tagging, chunking, NER, and semantic role labeling. Further interesting works on NER are (Piskorski et al., 2017) focusing on multilingual NER for slavic languages, and (Yadav and Bethard, 2018) presenting a broad survey of NER methods. In practice, a NER system can be crafted by fine-tuning a pre-trained LM, e.g., BERT (Devlin et al., 2018), RoBERTa (Liu et al., 2019), or DeBERTa (He et al., 2021). Nonetheless, these models are data hungry and need expensive GPUs during its training and inference. Further work has been directed at reducing their computational footprint, by performing, for example, knowledge distillation (Sanh et al., 2019).

Air traffic control communications typically carry structured information. A typical ATCo-pilot utterance consists of three major entities: The Callsign as plane-identifier, which is followed by the command and a value that specifies the command further. An example of the entities is shown in Figure 7. These three groups can be seen as 'named entities'. The *ATCO2-test-set corpus* provides transcription on the word level that assigns pieces of text to these specific classes. We developed a baseline system to extract such information from ASR utterances, as depicted in Figure 7. An early implementation of this system was covered in (Nigmatulina et al., 2022). However, these experiments were carried over private databases, so it is difficult to compare with our current results. That is why we base our experiments in (Nigmatulina et al., 2022), but we go beyond by open sourcing scripts to fine-tune a NER model with *ATCO2-test-set corpus*.

### 7.1.1 EXPERIMENTAL SETUP

Our experiments are carried out with *ATCO2-test-set corpus* only, for both, training and evaluation.[29] The main reason is that none of the public databases from Table 1 contain NER transcriptions. As a workaround, we implemented a simple k-fold cross-validation scheme. We define $K = 5$ folds, with a 70/10/20 ratio for train/dev/test subsets, respectively. We use ground truth ASR transcripts for training and testing NER.

First, we download a powerful pre-trained LM, BERT[30] (Devlin et al., 2018), from HuggingFace (Wolf et al., 2020; Lhoest et al., 2021). We append a linear layer with a dimension of 8 on top of the last layer of the BERT model.[31] The model is later fine-tuned on the NER task, with each Fold $K$ of the train splits. Each model is fine-tuned on an NVIDIA GeForce RTX 3090 for 10k steps. During experimentation, we use the same learning rate of $\gamma = 5e-5$ with a linear learning rate scheduler. Dropout (Srivastava et al., 2014) is set to

---

29. We provide in the GitHub repository the utterance IDs splits utilized for these experiments.
30. We use the pre-trained version of `bert-base-uncased` with 110 million parameters for all the experiments.
31. Following the Inside–outside–beginning (IOB) format, i.e., two outputs for each NER class.

Table 7: Different performance metrics for callsign, command and values classes of the NER system. Metrics reported for each of the 5-fold cross-validation scheme on *ATCO2-test-set corpus* with a `bert-base-uncased` model. @P, @R, and @F1 refer to precision, recall and F1-score, respectively. Numbers in **bold** refer to the top performance per column among folds. [†]mean score over the 5 folds.

| Fold | Callsign | | | Command | | | Values | | |
|------|------|------|------|------|------|------|------|------|------|
| | @P | @R | @F1 | @P | @R | @F1 | @P | @R | @F1 |
| 1 | 0.97 | 0.98 | 0.97 | 0.80 | 0.81 | 0.81 | 0.86 | 0.86 | 0.86 |
| 2 | 0.97 | 0.98 | 0.97 | **0.83** | **0.86** | **0.85** | 0.86 | 0.89 | 0.87 |
| 3 | 0.97 | 0.97 | 0.97 | 0.81 | 0.85 | 0.83 | **0.87** | 0.87 | 0.87 |
| 4 | **0.98** | **0.98** | 0.98 | 0.78 | 0.80 | 0.79 | 0.85 | **0.90** | 0.87 |
| 5 | 0.97 | 0.98 | **0.98** | 0.80 | 0.83 | 0.81 | **0.87** | 0.89 | **0.88** |
| AVG[†] | 0.97 | 0.98 | 0.97 | 0.80 | 0.83 | 0.82 | 0.86 | 0.88 | 0.87 |

$dp = 0.1$ for the attention and hidden layers, while Gaussian Error Linear Units (GELU) is used as activation function (Hendrycks and Gimpel, 2016). We also employ gradient norm clipping (Pascanu et al., 2013). We fine-tune each model with an effective batch size of 32 over 50 epochs with AdamW (Loshchilov and Hutter, 2019) optimizer ($\beta_1$=0.9, $\beta_2$=0.999, $\epsilon$=1e−8).

### 7.1.2 RESULTS

We report the results obtained from the 5-fold cross validation experiments. We split the results by tags, namely, callsign, command and values. For each of them, we report precision, recall and F1-scores in Table 7. We obtained an average of 0.97, 0.82 and 0.87 F1-score for callsign, commands and values. We observed that the command class was the most challenging among the three classes. We believe this is because commands contain extra complexity in comparison to callsigns and values. For example, in some cases the ATCos or pilots use several commands, or these are sometimes mixed in the same utterance. In contrary, callsigns follow a standard form, composed of an airline designator, numbers, and letters (spelled in ICAO phraseology). Values are composed of cardinal numbers and some standard words, e.g., 'flight level'. We also noted a significant irregularity in performance for the command class between the 5 folds (see column: Command in Table 7). For example, worse → best scenario on F1-score was 0.79 → 0.85, almost a six-point drop. A five-point drop is also seen in precision and recall. These results are seen when comparing fold 2 (best) against fold 4 (worst).

In conclusion, the results from Table 7 are the first official baseline for NER[32] on the *ATCO2-test-set corpus*. However, there is room for improvement. For instance, implementing semi-supervised learning or data augmentation should bring robustness and yield higher performance. Similarly, one can pretrain the LM directly on ATC text rather than standard

---

32. After extensive research, to authors' knowledge, this is the first official baseline on NER for air traffic control communications. We have not found any other work that is, both open-source and that targets NER.

English text, which should bring in additional benefits. We leave this line of research for future work.

Additional research towards named-entity recognition for air-traffic control communications with ATCO2 is carried in (Zuluaga-Gomez et al., 2023a). Here, the authors evaluate BERT (Devlin et al., 2018) and RoBERTa (Liu et al., 2019) models for NER.

## 7.2 Callsign Recognition and Understanding

The Named Entity Recognition system from §7.1 is capable to select words which form a callsign (i.e., highlight 'swiss two six eight nine'). However, *ICAO Callsign Extraction* produces the callsign directly in ICAO format (e.g., SWR2689), which is more useful for applications. This is not trivial because callsigns get commonly shortened, if the situation is obvious (e.g., 'swiss two six eight nine' → 'six eight nine', or 'swiss eight nine'). And the underlying ASR produces errors in its automatic transcripts.

In the project, we explored two approaches. In (Blatt et al., 2022), the ICAO callsign is retrieved by a BERT-based Encoder-Decoder neural network. This system directly takes outputs from an in-domain ASR system and extracts the ICAO

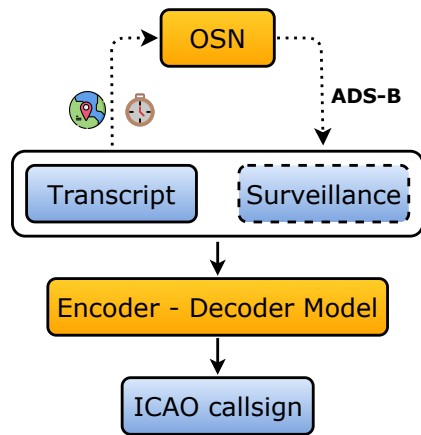

Figure 8: Proposed callsign recognition and understanding system. The dotted path marks the optional surveillance retrieval via OSN with the aid of the transcripts timestamp and VHF receiver location. Taken from Blatt et al. (2022).

callsign without relying on Named Entity Recognition as an intermediate step. The model uses a list of callsigns, i.e., context information to predict the callsign in ICAO format. The advantage of this sequence-to-sequence approach is, that it does not just select best callsign from the surveillance list, but it can also extract unknown callsigns, that are not present in the initial list. The overall approach is depicted in Figure 8.

The second approach (Nigmatulina et al., 2022) performs NER to extract the callsign within the sentence, which is later ranked by Levenshtein distance with the ones in the callsign list from the surveillance data. This approach always selects a callsign from the list. We showed that boosting callsigns with the combination of ASR and NLP methods eventually leads up to 53.7% of an absolute, or 60.4% of a relative, improvement in callsign recognition.

## 7.3 Speaker Role Detection

In NLP, text classification or sequence classification (SC) is a task that assigns a label or a class to a sequence of words (He et al., 2020; Zhou et al., 2015). The hypothesis is that

the words within the given text share a common role and meaning inside the sentence's grammatical structure. One of the most acknowledged forms of SC is sentiment analysis, which assigns a label like positive, negative, or neutral to a sequence of text embeddings[33] (Birjali et al., 2021). Nowadays, state-of-the-art SC systems are based on the well-known Transformer, e.g., BERT (Devlin et al., 2018) or RoBERTa (Liu et al., 2019). Akin to NER, SC is considered a downstream task operating on ASR output.

In ATC, the dialogues are built on top of a well-defined lexicon and dictionary, which follows a simple grammar. This standard phraseology has been defined by the ICAO (ICAO, 2020) to guarantee the safety and reduce miscommunications between the ATCos and pilots. In this work, we propose some baselines on the SC task aimed at detecting the speaker role from transcribed ATC communications (sentences). Our previous work on speaker role detection is covered in Zuluaga-Gomez et al. (2023d); Prasad et al. (2022a).

### 7.3.1 EXPERIMENTAL SETUP

The SC experiments are carried out in a very related manner to NER (see § 7.1.1). Specifically, we use the same model (`bert-base-uncased`), hyperparameters (e.g., number of epochs), optimizer, dropout rates and so on. However, here, we fine-tuned the model on the SC task rather than NER. We append a linear layer with a dimension of 4 (following the classes structure from Section 3.2 of (Nigmatulina et al., 2022)) on top of the last layer of the BERT model, i.e., a two-class classification model.

We employed LDC-ATCC[34] and UWB-ATCC[35] datasets (see Table 1) for fine-tuning and *ATCO2-test-set corpus* for testing. In LDC-ATCC and UWB-ATCC databases, speaker roles tags for each sample are marked in the original transcripts. And, we use ground truth ASR transcripts the evaluation. We create speaker-independent train/test splits based on the original databases. The split IDs for each subset are registered in the public GitHub repository of this paper.

### 7.3.2 RESULTS

We report the baseline results for speaker role detection in Table 8. Differently from NER, we only used *ATCO2-test-set corpus* for evaluation. We trained three models using different training datasets. From Table 8 we can see that pilots' communications are more challenging for our model in comparison to the ones from ATCos. For instance, in the model fine-tuned with LDC-ATCC corpus, there is a two-point drop in F1-scores for pilots, i.e., $0.79 \rightarrow 0.77$ F1-score. Similar behavior is seen in the model fine-tuned with UWB-ATCC corpus, i.e., a four-point drop in F1-scores, $0.86 \rightarrow 0.82$. However, models trained on the later show more robustness for both classes in comparison to the one trained with LDC-ATCC.

---

33. However, a sequence of acoustic embeddings can also be used for SC, e.g., emotion classification in raw speech (Purohit et al., 2022).

34. The Air Traffic Control Corpus (LDC-ATCC) corpus is public in: `https://catalog.ldc.upenn.edu/LDC94S14A`. It comprises recorded speech for use in the area of ASR for ATC. The audio data is composed of voice communication traffic between various controllers and pilots.

35. The UWB-ATCC corpus is released by the University of West Bohemia, and it can be downloaded for free in: `https://lindat.mff.cuni.cz/repository/xmlui/handle/11858/00-097C-0000-0001-CCA1-0`. The corpus contains recordings of communication between ATCos and pilots. The speech is manually transcribed and labeled with the speaker information, i.e., whether ATCo or pilot is speaking and when.

Table 8: Different performance metrics for the speaker role detection experiments. Metrics reported on *ATCO2-test-set corpus* with a `bert-base-uncased` model trained on the splits from Table 3. @P, @R, and @F1 refer to precision, recall and F1-score, respectively. Numbers in **bold** refer to the top performance per column.

| Training Corpus | ATCO | | | PILOT | | | AVG |
|---|---|---|---|---|---|---|---|
| | @P | @R | @F1 | @P | @R | @F1 | @F1 |
| LDC-ATCC | 0.87 | 0.73 | 0.79 | 0.70 | 0.86 | 0.77 | 0.78 |
| UWB-ATCC | 0.88 | **0.83** | **0.86** | **0.80** | 0.85 | 0.82 | **0.84** |
| LDC-ATCC + UWB-ATCC | **0.92** | 0.78 | 0.85 | 0.76 | **0.91** | **0.83** | **0.84** |

We also investigated the performance benefit of combining both datasets. For this experiment, we only obtained one point increase for the pilot class, while one point decrease for the ATCo class, both in comparison to the model trained on UWB-ATCC only. It is important to keep in mind that *ATCO2-test-set corpus* is a completely unseen dataset throughout all the experiments. We are convinced that integrating a small in-domain development set could boost the performances.

Additional research towards text-based speaker diarization and speaker role detection is carried in (Zuluaga-Gomez et al., 2023a). Here, the authors ablate SC for ATC with different models architectures, including BERT (Devlin et al., 2018), RoBERTa (Liu et al., 2019) and DeBERTa (He et al., 2021). Further, a text-based speaker diarization system is proposed with different in-domain ATC datasets.

## 8 Legal and privacy aspects for collection of ATC recordings

In order to safely distribute and make available the ATCo Corpus to the community, we took into account legal and ethical considerations as a prerequisite to distribute this content both, commercially and as open-source package. The main question we faced was to determine whether we could legally record and distribute ATC. To answer that question, we inquired into how legislation and regulations treat ATC (Rigault et al., 2022).

Our first hypothesis was that ATC would fall under the rules of Intellectual Property law which regulate how authors and companies can collect and use immaterial works such as recordings, where we believed ATC could fall into. To confirm this hypothesis, we aimed to find out whether ATC could be considered as material that could be protected by intellectual property legislation. Therefore, we performed a thorough study of the two major legal intellectual property systems of the United States and Europe. This study showed that due to the specific characteristics of ATC, such as phraseology and context, these conversations could not be protected as such as they do not meet the originality threshold for protection.

Then we moved on to another hypothesis. We thought that even if these conversations can not be protected as such, they might be protected as part of databases collected by either aircraft companies or Air Navigation Services Providers (ANSPs). As part of our investigations, we came into contact with some of these stakeholders, but none of them replied favorably to our requests. For example, the National Air Traffic Services which handles the airspace for the United Kingdom replied to us that they could not provide their

recordings unless mandated by a Court order. Moreover, the United Kingdom is one of the few countries that expressly prohibit recording ATC communications, as its legislation strictly prohibits the use of unlicensed recording apparatus[36].

Other countries have a more lenient policy towards access and recording of ATC. Indeed, during our research we found out that ATC recorded in the US could be accessed on demand by formulating a request to the Federal Aviation Administration (FAA, hereafter). This is made possible by the use of the Freedom of Information Act (FOIA) that compels US administrations to make available certain types of information collected by these administrations during their operations. In the specific case of the FAA, Freedom of information regulations states that radio and computer data can be obtained upon request as stated in Chapter 4 Section 4-8-2 of the Facility Operation and Administration Order [37]. Nevertheless, during our exchanges with the FAA we found out that requests for audio files needed to concern the last 45 days as required by Chapter 12, section 2 Article 12-2-2 of the same order and had to be specific to an airport in order to be processed adequately by the administration. Future work may include the drafting of such a request to confirm the reality of those conversations.

For the airports based in Europe, we based our collection process on the existence of the Open Data Directive, formerly known as the Public Sector Directive, whose goal is to allow reuse and redistribution of data collected by public services, as we found out, many ANSPs are either State-owned or run by state administration for obvious security reasons. Therefore, we made a request in France to access data collected by the administration in charge of Air Traffic Control. We based our request on the provisions of French Law allowing to request the access to data produced by this administration. Regarding ATC, our request went up to the Commission d'Accès aux Documents Administratifs (Commission for access to administrative documents). This Commission ruled that the Direction Générale de l'Aviation Civile (DGAC) did not have to fulfill our request for data since they could not differentiate between civilian and military aircraft and that the recordings could leave the identification of speakers. However, following an *a contrario* interpretation, it can be assumed that since our project focussed on civilian aircraft and that we ensured the anonymization of the conversations before making them available, we could pursue data collection.

This previous ruling raised our concerns regarding the compliance of the project especially with the regulations related to protection of personal data, especially the EU General Data Protection Regulation (GDPR). GDPR is the main text regulating the collection and distribution of databases containing personal data. In the case of ATC, the speech data contains voiceprints of the pilots and ATCos, which can be used as a mean of identification through speaker identification techniques. Thus, further precautions should be adopted in order to be able to collect this type of data. However, GDPR allow the collection of speech data when made in relation to reasons of substantive public interest, which we found applicable in our case since the project is aimed at enhancing airspace security.

---

36. Further information in the following url: `https://www.legislation.gov.uk/ukpga/2006/36/section/48`

37. JO 7210.3CC—Facility Operation and Administration available at `https://www.faa.gov/air_traffic/publications/atpubs/foa_html`

## 9 Conclusions and Future Work

This article introduces, the *ATCO2 corpus*, a set of three corpora for research on robust automatic speech recognition and natural language understanding of air traffic control communications. In ATCO2, we have successfully created and deployed an operating pipeline for collecting, pre-processing and automatically transcribing ATC audio data. During the data collection period, we mostly relied on a worldwide community of volunteers that acted as 'data feeders'. Then, a community of 'data annotators' employed the SpokenData transcription platform to generate gold transcriptions of a small portion of the collected data. The platform is up and running, and it is reachable on `https://www.spokendata.com/atco2`.

The *ATCO2 corpus* is divided in *ATCO2-PL-set corpus* and *ATCO2-test-set corpus*. The former contains more than 5000 hours of automatically transcribed ATC speech data, spanning more than ten airports in different continents (Table 4). While the latter, *ATCO2-test-set corpus*, contains gold transcriptions of 4 hours of ATC speech. A subset called *ATCO2-test-set-1h corpus* is offered for free in `https://www.atco2.org/data`. To the authors' knowledge, this is the first public release of a large-scale database for research in the area of air traffic control communications.

In addition, we also cover baselines (our source code for data preparation and to replicate the NLU baselines is stored in the following public GitHub repository `https://github.com/idiap/atco2-corpus`.) over three different key tasks in the area of ATC. The first one is related to robust ASR, while the next two are about to NLU of ATC communications.

(1) We demonstrated that training an ASR system solely on *ATCO2-PL-set corpus* reaches competitive WERs on both, public and private databases (see Table 6). This is important because *ATCO2-PL-set corpus* is purely composed of pseudo labels generated by ATCO2 project seed ASR system. This can be the starting point for many researchers and companies worldwide that would like to use our corpora for testing and training robust ASR systems for ATC. Furthermore, our results go in line to systems like Whisper (Radford et al., 2023). Whisper proved that weekly-supervised labels helps ASR at scale, when hundreds of thousand of audio data are used during training.

(2) We demonstrated that as much as 3000 utterances are needed to train and evaluate a BERT-based Named Entity Recognition system for ATC communications. This system is capable of detecting callsigns, commands, and values from the textual inputs. This NLU task is of special interest to the ATC community because this high-level information can be used to assist ATCos in order to reduce their overall workload.

(3) Similarly, we developed a simple yet efficient BERT-based module that performs speaker role detection from textual inputs. It can say whether an input ATC utterance was spoken by an ATCo or a pilot.

Finally, we believe that the 'lessons learned' in ATCO2 project and its recipe for collecting and pre-transcribing large-scale audio databases can be easily transferred to other applications, where data scarcity is a latent problem.

### Nomenclature

- AI: Artificial Intelligence
- AM: Acoustic Model

- ASR: Automatic Speech Recognition
- ATC: Air Traffic Control
- ATM: Air Traffic Management
- ATCo: Air Traffic Controller
- ATCC: Air Traffic Control Corpus
- CTC: Connectionist Temporal Classification
- Conformer: Convolution-augmented Transformer
- dB: Decibel
- DNN: Deep Neural Networks
- ELDA: European Language Resources Association
- End2End: End-To-End
- FST: Finite State Transducer
- GELU: Gaussian Error Linear Units
- GDPR: General Data Protection Regulation
- IOB: Inside–outside–beginning
- ICAO: International Civil Aviation Organization
- LM: Language Model
- LF-MMI: Lattice-Free Maximum Mutual Information
- ML: Machine Learning
- NER: Named Entity Recognition
- NLP: Natural Language Processing
- NLU: Natural Language Understanding
- RBE: Read-back Error
- RBED: Read-back Error Detection
- SNR: Signal-To-Noise
- SLU: Spoken Language Understanding
- VAD: Voice Activity Detection
- VHF: Very-High Frequency
- WFST: Weighted Finite State Transducer
- WER: Word Error Rate

## Broader Impact Statement

Our research on advancing automatic speech recognition (ASR) and natural language understanding (NLU) for air traffic control (ATC) communications carries both potential benefits and ethical considerations. The development of the *ATCO2 corpus* and the associated open-source resources marks a significant push forward in addressing the low-resource nature of the ATC domain. This initiative contributes to broader advancements in robust ASR and NLU applications to other applications. The *ATCO2 corpus* presents opportunities for innovation, fostering safer and more efficient air travel, once is AI-based systems are integrated in real-life ATC operations. We acknowledge the importance of ethical considerations in deploying AI technologies in safety-critical domains. In this work we stress the need for rigorous testing, validation, and ongoing monitoring to ensure the responsible integration of these technologies into the ATC operation rooms. Therefore:

- **Extensive testing on diverse datasets**: the ATCO2 corpora, with its large-scale collection of ATC audio and pseudo labels from multiple airports (i.e., countries), stand as an ideal testbed across various acoustic conditions, accents, and local conventions. Thus, we provide labels for this regard;

- **Regular validation of pseudo labeled data**: the usage of pseudo labels for ASR and NLP tasks requires regular verification by annotators to ensure that audio containing new information (e.g., new airline designators) adheres to the standard phraseology;

- **Continuous performance monitoring**: a proper integration of ASR and NLP tools into the ATC operation rooms need regular checks against manually annotated data to detect any degradation in performance over time;

- **Gradual integration into operation rooms**: an initial deployment of ASR and NLP tools should be in a supportive role, with ATCos maintaining final decision-making authority.

## Acknowledgments and Disclosure of Funding

This paper introduces the *ATCO2 Corpus* derived from a joint collaborative project supported by Clean Sky 2 Joint Undertaking (JU) under grant Agreement No. 864702 - ATCO2 (Automatic collection and processing of voice data from air-traffic communications).

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

## Appendix A. Automatic Transcription Engine

This appendix describes in details how we collected the audio and metadata that brought to live the *ATCO2 corpus*. We mainly rely on the automatic transcription engine, described in more details in Section 4.3. The automatic transcription engine is implemented as a scalable cloud service. It communicates with other services (or partners) using APIs. This service is designed to process large flows of data produced by data feeders.[38]

The data is pushed to this service by OSN[39] servers by calling an API request and providing a job setting JSON file. After the request is accepted, settings parameters are processed and the job is stored in an internal queue for processing. The user (in this case, OSN) may have an ability to tweak the settings and to affect the processing pipeline and the result. Namely:

- Audio input format choices;

- Rejection threshold for too long audios;

- Rejection threshold for too short audios;

- Rejection threshold for too noisy audios;

- Rejection threshold for non-English audios;

- Switching the language of automatic speech recognizer.

Most of these are actually disabled due to security reasons (not to interrupt the processing pipeline), but may be easily enabled on the fly if needed. The overall data flow model is described in Figure 9. Any new job (request for a full automatic transcription of recording) accepted via API on the SpokenData[40] side is processed by a master processing node. The job is enqueued into a workload manager queue. Once there is a free processing slot, the job is submitted to a processing server, or worker. The master processing node then informs the OSN server about the state of the job by calling a callback.

---

38. Enthusiasts that act as 'feeders' of ATC speech and contextual ATC data (surveillance). See Section 4.1.

39. OpenSky Network: `https://opensky-network.org/`.

40. Industrial partner: `https://www.spokendata.com/atco2`.

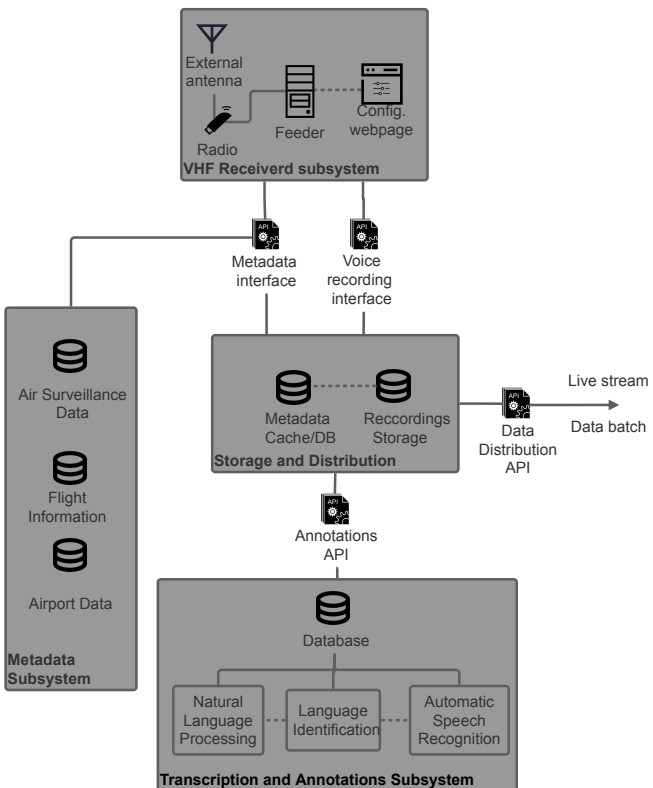

Figure 9: Overall ATCO2 communication schema.

## Appendix B. Unification of transcripts

This appendix conveys our main results of transcripts unification and lexicon formatting. Note that the description below is related to the databases employed to train the seed ASR engine used during the pre-transcription process, described in Section 4.3. Special attention was devoted to the unification of words from the radiotelephony alphabet and numbers (see *ICAO alphabet* column from Table 9). Note that we map the word 'niner' → 'nine', and in the pronunciation lexicon, we allow the word 'nine' to be pronounced as `'n ay1 n er0'` (phoneme-based format). Also, some standard expressions can be written as two words or as a single word. For some of the frequent ones, we selected one version that is used systematically (see *Common expressions* column from Table 9). We also rectify some airline designators that are part of the callsigns uttered by the ATCos and pilots (see *Airline designator* column from Table 9).

We derive a table of mapping rules by extracting insights from a diverse list of airline designators. In total, we have a list of 5.4k airline designators, out of these, there are 1.8k multi-word airline designators. The airline designators ligatured by underscore are easier to be produced by the 'speech-to-text' system as the tokens are longer, and there is also less uncertainty to be modelled by the language model. Finally, we pay extra attention to the transcripts generated by the ATCO2 community of volunteers. Like in any other human input, there might be typos or other types of transcription errors. It is necessary to at least revise the transcripts by the 2nd round of human inspection, where the errors are ideally fixed.

Table 9: Normalization rules applied for transcription.

| Unification of transcripts | | |
|---|---|---|
| **ICAO alphabet** | **Common expressions** | **Airline designators** |
| alpha → alfa | take off → takeoff | qatar → qatari |
| charly → charlie | call sign → callsign | turkey → turkish |
| juliet → juliett | readback → read back | air france → airfrans |
| oskar → oscar | flightlevel → flight level | norshuttle → nor shuttle |
| xray → x-ray | stand by → standby | airvan → air van |
| zoulou → zulu | start up → startup | rynair → ryanair |
| whisky → whiskey | goodbye → good bye | airbaltic → air_baltic |
| tripple → triple | clear for → cleared for | air berlin → air_berlin |
| niner → nine | lineup → line up | air canada → air_canada |
| 0 → zero | clear for → cleared for | air china → air_china |
| 1 → one | turnright → turn right | air europe → air_europe |
| 2 → two | oclock → o'clock | jet stream → jet_stream |
| 3 → three | o clock → o'clock | jetstream → jet_stream |
| 4 → four | push back → pushback | k l m → k_l_m |
| 5 → five | descent direct → descend direct | klm → k_l_m |
| 6 → six | goodbye → good bye | korean air → korean_air |
| 7 → seven | goodday → good day | koreanair → korean_air |
| 8 → eight | turbulance → turbulence | wizzair → wizz_air |
| 9 → nine | til → till | top_jet → topjet |

## Appendix C. How a Sample From ATCO2 corpora Looks Like?

Example of human transcriptions for a recording of *ATCO2-test-set corpus* in XML format. This file encodes most of the metadata. If more than one ** is present, it means there are two or more utterances in the recording:

```xml
<?xml version="1.0" encoding="utf-8"?>
<data>
        
                <start>0</start>
                <end>2.93</end>
                <speaker>B</speaker>
                <speaker_label>pilot</speaker_label>
                <text>[unk] [#callsign]Quebec Lima[/#callsign] [#command]
    confirm cleared for ILS[/#command] [unk]</text>
                <tags>
                        <correct>0</correct>
                        <correct_transcript>1</correct_transcript>
                        <correct_tagging>0</correct_tagging>
                        <non_english>0</non_english>
                </tags>
        
        
                <start>2.99</start>
```

```
18                   <end>10.45</end>
19                   <speaker>A</speaker>
20                   <speaker_label>ATCo approach</speaker_label>
21                   <text>[unk] [#callsign]Quebec Lima[/#callsign] [#command]
     affirm cleared ILS approach[/#command] [#value]runway one four[/#value]
     [#command]if you go around[/#command] [#value]runway one four[/#value]
     [#command]report in localizer established[/#command]</text>
22                   <tags>
23                           <correct>0</correct>
24                           <correct_transcript>1</correct_transcript>
25                           <correct_tagging>0</correct_tagging>
26                           <non_english>0</non_english>
27                   </tags>
28           
29  </data>
```

Listing 1: XML tagged example from *ATCO2-test-set corpus.* This example contains two recordings separated by the ** tag.

Basic details from the previous XML tagged segment:

-  ... : one sample of data. One recording may have one or more segments;

- <start> ... </end>: timing information with speech activity by the speakers;

- <speaker> ... </speaker>: speaker information to identify whether the segment is from an ATCo or pilot. Unknown cases are tagged with <UNK>

- <text> ... </text>: ground truth transcripts with high-level entities transcriptions (callsigns, commands and values). Not all the segments contains these transcriptions.

- <tags> ... </tags>: extra metadata.

