# OpenReview forum: "ATCO2 corpus: A Large-Scale Dataset for Research on Automatic Speech Recognition and Natural Language Understanding of Air Traffic Control Communications"
_DMLR — Accepted by DMLR_

### Review · Reviewer_raVC · 2024-03-20

**Recommendation:** 4
**Confidence:** 1

**Summary Of Contributions:**

This submission introduces a publicly available corpus called ATCO2 which aims to address several challenges in research on air traffic control (ATC) communication. First, existing corpora focus on speech recognition (ASR) while ATCO2 includes data for other tasks like named entity recognition (NER). Second, ATC communication data is expensive to annotate due to its complexity. ATCO2 uses a combination of automatic pre-annotation and human correction to reduce costs. Third, most ATC corpora come from a limited number of airports, hindering the development of generalizable models. ATCO2 includes data from various airports to address this domain shift problem. Finally, the noisiness of real-world ATC recordings makes them a valuable resource for training robust speech understanding systems. Overall, the ATCO2 corpus would be a valuable resource for researchers and practitioners working on automatic speech recognition and natural language understanding for ATC communications. I recommend to accept this submission.

**Strengths:**

1. Data processing pipeline carefully considers all the necessary components: (1) speech pre-processing tools (segmentation, volume
adjustment and discarding noisy recordings), (2) diarization (split audio per speaker), (3) ASR, (4) English language detection (ELD), (5) speaker role detection (SRD) e.g., ATCo or pilot, and (6) labeling of callsigns, commands and values with named entity recognition (NER).
2. In particular, it is highly critical that this dataset goes beyond simple ASR for the task -- mimicking real-world scenarios.
3. The paper is very well-written, and the dataset + code is well-organized.

**Audience:**

Yes

**Broader Impact Concerns:**

Authors acknowledge the importance of ethical considerations in deploying AI-based technologies in safety critical domains such as the ATC domain. I would love to see another paragraph or two on their recommendations related to "rigorous testing, validation, and ongoing monitoring to ensure responsible integration...". That would be immensely helpful for guiding the researchers and practitioners who would be working on those things.

**Claims And Evidence:**

I am not very familiar with the ATC field -- but from what I can understand from the paper, claims made are convincing.

**Datasets And Benchmarks:**

Github repo is very well-organized, and it provides all the necessary pointers to the data and how to use it. I have no concerns here.

**Extended Submissions:**

My understanding is that this submission adds transcribed audio as the ATCO2-test-set -- extending the previously published ATCO2 corpora (see Table 1 in the submission). I would appreciate it if authors could make the additional delta compared to their previous work on ATCO2 in Section 2. DMLR clearly states that extended versions should contain a minimum of 30% additional content compared to their prior versions. I will leave it to AE to confirm if that test set would constitute an additional 30%. Please clarify if there is a misunderstanding here.

**Limitations:**

I do not have major concerns about this work from an engineering perspective. However, here are the mild concerns I have regarding the limitations:

1. Volunteer-driven data collection: The quality and consistency of the data may vary due to the reliance on volunteers with potentially diverse levels of experience and equipment.
2. Potential geographic bias: Data might be concentrated in regions with a higher density of aviation enthusiasts, leading to a skewed representation of air traffic patterns.
3. Regulatory restrictions: Legal limitations in some countries could hinder data collection, and discarding noisy recordings might result in the loss of potentially usable data.

**Requested Changes:**

1. Broader Impact Recommendations: Expand the discussion on "rigorous testing, validation, and ongoing monitoring" for responsible integration of AI in safety-critical domains like ATC. This would provide valuable guidance for researchers and practitioners.

2. Extended Submission Clarification: In Section 2, clearly explain the additional content (transcribed audio as ATCO2-test-set) compared to the previously published ATCO2 corpora (referencing Table 1). Verify if this addition meets the DMLR requirement of at least 30% more content for extended submissions.

**Strengths And Weaknesses:**

This well-written paper presents a well-structured data processing pipeline that thoughtfully incorporates segmentation, speaker diarization, automatic speech recognition (ASR), language detection, speaker role detection, and named entity recognition. The emphasis on going beyond simple ASR for real-world applicability is particularly valuable, along with the clear writing and organization of dataset and code. However, the reliance on volunteer-driven data collection raises mild concerns about potential variations in quality and consistency. Additionally, geographic bias and regulatory restrictions may limit the dataset's representativeness and scope.

---

### Review · Reviewer_Hg7S · 2024-05-23

**Recommendation:** 3
**Confidence:** 1

**Summary Of Contributions:**

This work develops a large-scale dataset for ASR and NLU in the Air Traffic Control (ATC) communication domain. The dataset could potentially be useful for future studies, especially as ATC is a low-resource domain. The paper also presented several baselines of ASR and NLU on the new dataset.

**Strengths:**

* This work presents a new dataset for the ATC domain, which is large-scale and systematically developed. It could be helpful for future research in this domain, as well as future works studying domain shifts of ASR or NLP.
* Results of several baselines are also presented in the paper.
* The paper is very detailed. The dataset is publicly available already. There are also links for users to contribute to the dataset.

**Audience:**

Yes

**Broader Impact Concerns:**

I think the authors also need to discuss the potential risks associated with the pseudo-labeled dataset,  and potentially malicious feeders or annotators.

**Claims And Evidence:**

More evidence is needed for part of the paper. See above.

**Datasets And Benchmarks:**

Yes.

**Extended Submissions:**

It has been mentioned that "ATCO2 corpora has been used for developing virtual simulation pilots in (Prasad et al., 2022b) which was later extended in (Zuluaga-Gomez et al., 2023b)". It is unclear how much of this paper is from earlier versions.

**Limitations:**

See "Requested Changes" above.

**Requested Changes:**

* Evaluating the quality of the pseudo-labeled dataset.
* Adding empirical results which compare the performance of models trained on this new dataset v.s. previous datasets.
* Adding more explanation on how future annotations are added, how volunteers are obtained, how efficient it is to get volunteers, how the annotators are paid, etc.
* Adding empirical results or explanations on validating data from data feeders.

**Strengths And Weaknesses:**

Strengths:
* This work presents a new dataset for the ATC domain, which is large-scale and systematically developed. It could be helpful for future research in this domain, as well as future works studying domain shifts of ASR or NLP.
* Results of several baselines are also presented in the paper.
* The paper is very detailed. The dataset is publicly available already. There are also links for users to contribute to the dataset.

Weaknesses:
* Currently, the audios are pseudo transcribed. Therefore, there may be concerns on the quality of the dataset.
* It is not very clear how training with the new dataset helps on the downstream tasks, compared to training on existing dataset only (especially the dataset is pseudo-labeled).
* It is not very clear how future annotations will be added. The paper has mentioned that "we relied on both the volunteers and paid transcribers". It is unclear why some annotators are paid, and some are not. It is unclear how annoators can be obtained, especially this project doesn't pay the volunteers.
* It is unclear how data from data feeders are validated. What if someone uploads low-quality or even malicious data?

---

### Review · Reviewer_e4C8 · 2024-10-17

**Recommendation:** 3
**Confidence:** 2

**Summary Of Contributions:**

This paper introduces a novel corpus ATCO2, which contains audio (pseudo and manually transcribed) and radar data, for research on incorporating data-driven AI systems such as ASR and NLU to ATC dialogue communications (i.e., multi-speaker, multi-turn conversations) and management. These technologies are claimed to be able to help to reduce ARCo's workload. The author split the corpus into three subsets for different tasks. This paper also provides baselines using the invented dataset.

**Strengths:**

1. This is the first corpus that conveys ATC data with parallel labels for ASR and spoken-based tasks, including named entity recognition (NER), slot filling (SF), and sequence classification
 2. The collection and processing pipeline developed for ATCO2 can be generalised to different domains. E.g., call-centers conversations or medical recordings.

**Audience:**

Yes

**Claims And Evidence:**

The claims seem accurate with convincing evidence.

**Datasets And Benchmarks:**

This paper provides data collection protocol, methodologies for the transcription process, pre- and post-processing steps undertaken during the transcription process,  a review of data statics and baselines on ASR and NLP using the collected dataset.  The main legal and ethical implications of ATC data collection are also provided.

**Extended Submissions:**

Not applicable

**Limitations:**

1. The authors mentioned that "A significant bottleneck in the pipeline is the significant latency arising from an ATCo...". However, the significant latency and how can the proposed technologies solve the problem was not mentioned afterwards.
2. The authors mentioned accented speech cases in ATC but how to deal with this challenge was not explained in this paper.
3. I noticed that the authors used a SAD tool to remove non-speech parts, and I am wondering how the authors measured the performance of SAD performance.
 4. The ratio of the ATCO2-test-set and ATCO2-PL-set for training is around 1000:1. Is there any reason that the authors chose to split the test so little?
5. As ATCO2-PL-set has around 3000h which is not really low-resource and should be enough for an E2E model training. Why the author didn't consider to train an E2E system such as Conformer as it has been shown to outperform the Hybrid systems?

**Requested Changes:**

1. In the abstract, the "disadvantages" in "we make several contributions aiming at overcoming these disadvantages" are not clear before this statement.
2. Add descriptions to the different tasks you introduced when describing the pipeline developed in ATCO2 project, and make the connection or highlight parts of this work clear.
3. Add detailed descriptions for different terminologies in the figure  for the caption of Fig,2
4. "Our strategy to mitigate the out-of-vocabulary problem is based on enriching the lexicon as much as possible as part of the data preparation", how many OOV words in total that were added in the lexicon manually for ATCO2?

**Strengths And Weaknesses:**

Strengths:
1. The dataset is novel with transcriptions.
 2. The pipeline is generalised to other domains.
3. Four big challenges in ATC were answered by introducing the ATCO2 corpus.

Weakness
1. The paper mentioned several different tasks (e.g., NER, ELD) when giving an overview of the data processing pipeline. However, what those tasks are, the reason why those tasks are there and what are the connections between those tasks to the purpose of this work is confusing and should be clearly described.
2. The challenges are not clearly explained.